# Defective ventral neurogenesis due to midfetal *Chd8* mutation drives autistic-like behavior in mice

Kenta Nitahara[1,2,3,6], Atsuki Kawamura ✉[2,6], Ayumu Tashiro[2], Tomoya Iwasaki[2], Shin-Ichi Horike[4], Jumpei Terakawa[4], Takiko Daikoku[4], Koichi Higashi[5], Ken Kurokawa[5], Kiyoko Kato[3] & Masaaki Nishiyama[1,2] ✉

Autism spectrum disorder (ASD) is a common neurodevelopmental condition characterized by behavioral abnormalities. Although mouse models have been widely adopted to recapitulate the pathology of ASD, the identification of specific neural abnormalities responsible for autistic-like behavior has remained challenging. Here we provide insight into this causal relation by identifying the critical period and cell type responsible for the development of such behavior in ASD model mice with a *Chd8* mutation. We find that *Chd8* mutation induced at embryonic day 14.5 gives rise to ASD-like behavioral phenotypes, including abnormal social interaction and increased anxiety-like behavior, as well as to accelerated cell-cycle exit and differentiation in ventral progenitor cells. Restoration of *Chd8* expression in ventral progenitor cells ameliorates both the behavioral phenotypes and aberrant ventral differentiation in *Chd8* mutant mice. Our findings indicate that *Chd8* mutation during the midfetal period—in particular, in ventral progenitor cells—contributes to the development of autistic-like behavior.

Autism spectrum disorder (ASD) has emerged as an important socio-medical issue because of its high prevalence and distinctive symptoms[1-3]. ASD is characterized primarily by its behavioral deficits, including impaired social interaction and a pattern of restrictive, repetitive behaviors[2,3]. Given the difficulty in obtaining brain specimens of human patients, mouse models have become important research tools for studies of the pathology of neurodevelopmental disorders, including ASD[4-6]. Many mouse models of ASD have been developed and found to recapitulate neural abnormalities and behavioral symptoms of humans with ASD[7-9]. However, despite the identification of various alterations in neurogenesis in mouse ASD models, the primary neurodevelopmental phase and specific alterations responsible for the development of behavioral abnormalities have

remained poorly understood. Such knowledge is key to providing insight into ASD pathogenesis as well as to the development of new therapies.

Recent large-scale exome sequencing studies have revealed that genes related to chromatin regulation are frequently mutated in individuals with ASD[10-13]. The gene encoding chromodomain helicase DNA-binding protein 8 (CHD8), one of the major chromatin remodelers, has thus been identified as one of the most frequently mutated genes in ASD[10-13]. Whereas homozygous deletion of *Chd8* in mice results in early embryonic death[14], *Chd8* heterozygous mutant mice manifest behavioral abnormalities, macrocephaly, and gastrointestinal disturbances that resemble characteristics of ASD patients with *CHD8* mutations[14-18]. CHD8, similar to other chromatin remodelers, is widely expressed in

[1]Social Brain Development Research Unit, Next Generation Medical Development Research Core, Institute for Frontier Science Initiative, Kanazawa University, Kanazawa, Japan. [2]Department of Histology and Cellular Biology, Graduate School of Medical Sciences, Kanazawa University, Kanazawa, Japan. [3]Department of Gynecology and Obstetrics, Graduate School of Medical Sciences, Kyushu University, Fukuoka, Japan. [4]Research Center for Experimental Modeling of Human Disease, Kanazawa University, Kanazawa, Japan. [5]Genome Evolution Laboratory, National Institute of Genetics, Mishima, Japan. [6]These authors contributed equally: Kenta Nitahara, Atsuki Kawamura. ✉e-mail: a.kawamura@berkeley.edu; nishiyam@staff.kanazawa-u.ac.jp

various cell types and regulates gene expression essential for the development of both neural and nonneural organs[14,19,20]. In the cerebral cortex, CHD8 sustains the function of pyramidal neurons, with its deficiency resulting in structural disorganization of cortical layers and macrocephaly[15,21,22]. CHD8 also plays an important role in the proliferation and differentiation of progenitor cells for γ-aminobutyric acid–positive (GABAergic) interneurons[23,24] as well as of those for oligodendrocytes[25–27] and astrocytes[28]. Outside of the nervous system, CHD8 deficiency is associated with gastrointestinal symptoms[17,18], infertility[20], and slenderness[19]. Among the various neural and non-neural alterations that have been detected in mice with *Chd8* mutations, however, it has remained unclear which specific changes are primarily responsible for autistic-like behavior.

During neural development, the proliferation and differentiation of the various neural cell populations are precisely regulated in a spatiotemporally specific manner[29,30]. Disruption of this spatio-temporal harmony in neurogenesis is implicated in the development of ASD[31]. Elucidation of the pathology of ASD will require pinpointing when (critical period) and in which cell type the primary neural changes occur. Here we show that induction of *Chd8* mutation during the midfetal stage leads to behavioral abnormalities, accompanied by accelerated cell-cycle exit and differentiation of ventral progenitor cells. We further demonstrate that restoration of *Chd8* expression in these cells ameliorates both neural and behavioral phenotypes. These findings establish a spatiotemporal framework linking disrupted ventral neurogenesis to autistic-like behavior.

## Results

### Identification of the critical period for development of autistic-like behavior due to *Chd8* heterozygous mutation

Given that CHD8 is broadly expressed in both neural and nonneural tissues[20], we first investigated whether neural cell–specific heterozygous mutation of *Chd8* would recapitulate the phenotypes observed in global *Chd8* heterozygous mutants. For this purpose, we generated *Nestin-Cre/Chd8*+/F mice to induce neural stem cell–specific *Chd8* heterozygous mutation[32]. Neural stem cell–specific heterozygous mutation of *Chd8* resulted in macrocephaly as well as behavioral abnormalities, including increased anxiety-like behavior, reduced locomotion, and increased contact time in both female and male mice (Supplementary Fig. 1a–l); of note, no aggressive interactions were observed during the direct social-interaction test. On the other hand, no significant difference was observed with regard to cognitive flexibility, motor stereotypies (self-grooming test and circling behavior), sensory processing, or the sociability index and social-novelty index in three-chamber tests (Supplementary Fig. 1m–o and Supplementary Fig. 2). Overall, these phenotypes resembled those for global heterozygous mutation of *Chd8*[14–18].

We therefore next attempted to identify the critical developmental time point at which *Chd8* mutation gives rise to behavioral phenotypes. To elicit *Chd8* mutation during embryonic or postnatal periods, we adopted *Nestin-CreER*T2 and *CAG-CreER* systems, respectively, which allow the induction of Cre-mediated recombination at a specific time point by tamoxifen administration[33,34]. Given that the onset of Nestin-Cre activity has been shown to occur between embryonic day (E) 9.5 and E11.5[35–37], we induced *Chd8* mutation by tamoxifen administration at E11.5 (early fetal stage), E14.5 (midfetal stage), E17.5 (late fetal stage), or postnatal day (P) 1 to P3 (postnatal stage) (Fig. 1a). Mice exposed to tamoxifen in utero were delivered by cesarean section and fostered, given that tamoxifen-treated dams can experience difficulties in vaginal delivery[38]. Whereas induction of *Chd8* heterozygous mutation at E17.5 or later was not associated with behavioral abnormalities, such induction at E14.5 or earlier resulted in abnormalities in social contact in the reciprocal social-interaction test (Fig. 1b, Supplementary Fig. 3c, and Supplementary Fig. 4c, d) as well as in increased anxiety-like behavior in the light-dark transition, elevated

plus-maze, or open-field tests (Fig. 1c–h, Supplementary Fig. 3d, e, and Supplementary Fig. 4e–j), without the development of macrocephaly (Supplementary Fig. 3a, b and Supplementary Fig. 4a, b). We confirmed that the procedures of cesarean delivery and fostering alone did not induce behavioral changes in *Nestin-CreER*T2/*Chd8*+/F mice (Supplementary Fig. 5). These findings indicated that midfetal induction of *Chd8* heterozygous mutation is critical for the development of behavioral abnormalities.

### Lineage tracing and scRNA-seq analysis reveal cell type–specific gene expression changes due to midfetal induction of *Chd8* heterozygous mutation

We hypothesized that neural alterations observed during the critical period might underlie the development of autistic-like behavior. We therefore examined changes in gene expression associated with a *Chd8* heterozygous mutation at E14.5. To trace cells in which Cre-mediated recombination had occurred, we crossed *Nestin-CreER*T2 mice with mice harboring a tdTomato reporter gene with a CAG promoter and loxP-flanked stop (LSL) cassette at the *Rosa26* locus (*Rosa26*+/CAG-LSL-tdTomato, hereafter referred to as *Rosa26-tdTomato*)[39]. We compared tdTomato expression patterns for the brain of adult *Nestin-CreER*T2/*Rosa26-tdTomato* mice after tamoxifen administration at E11.5, E14.5, or E17.5 (Fig. 2a and Supplementary Fig. 6a). For mice in which tamoxifen was administered at E11.5, tdTomato was highly expressed in the forebrain, including the olfactory bulb, cortex, striatum, hippocampus, and thalamus (Fig. 2a, b). For those in which tamoxifen was administered at E14.5, a relatively high level of tdTomato expression was apparent in the telencephalon—including the olfactory bulb, cerebral cortex, striatum, and hippocampus—but not in the thalamus, whereas the expression persisted in the olfactory bulb, upper-layer cortex, and hippocampus for mice in which tamoxifen was administered at E17.5 (Fig. 2a, b). Given our identification of the critical period, we inferred that the cerebral cortex and striatum—in both of which tdTomato expression was pronounced for mice treated with tamoxifen at E14.5 but was downregulated for those treated at E17.5—may contribute to the development of autistic-like behavior.

To characterize tdTomato-positive cells and investigate the effects of *Chd8* mutation on neural development and function, we isolated these cells by fluorescence-activated cell sorting (FACS) from the telencephalon of P5 mice that had been treated with tamoxifen at E14.5 and then subjected the sorted cells to single-cell RNA-sequencing (scRNA-seq) analysis (Fig. 2c). After size-gating and removal of dead cells, about one-third of telencephalon cells were counted as tdTomato positive for both *Nestin-CreER*T2/*Rosa26-tdTomato/Chd8*+/+ (control) and *Nestin-CreER*T2/*Rosa26-tdTomato/Chd8*+/F (mutant) samples (Fig. 2d and Supplementary Fig. 6b, c). After quality control, scRNA-seq data processing yielded 8681 control cells and 8940 mutant cells for downstream analysis. We identified 12 molecularly distinct cell clusters by dimension reduction with uniform manifold approximation and projection (UMAP) and characterized these clusters on the basis of the expression of marker genes (Fig. 2e, f). Among these clusters, the proportions of excitatory neurons and microglia were higher for the control cells, whereas the proportions of endothelial cells, oligodendrocyte progenitor cells (OPCs), and oligodendrocytes were higher for the mutant cells (Fig. 2g, h). We then analyzed differential gene expression due to the *Chd8* heterozygous mutation induced during the critical period. Whereas many genes related to various biological processes were differentially expressed in the different clusters (Supplementary Fig. 6d, e and Supplementary Data 1), gene set enrichment analysis (GSEA) revealed substantial alterations in cell type–specific gene expression in the clusters corresponding to glutamatergic neurons, GABAergic neurons, and oligodendrocyte-lineage cells (Fig. 2i and Supplementary Data 2). Such gene expression was mostly downregulated in the mutant cells of these cell categories, but the gene sets

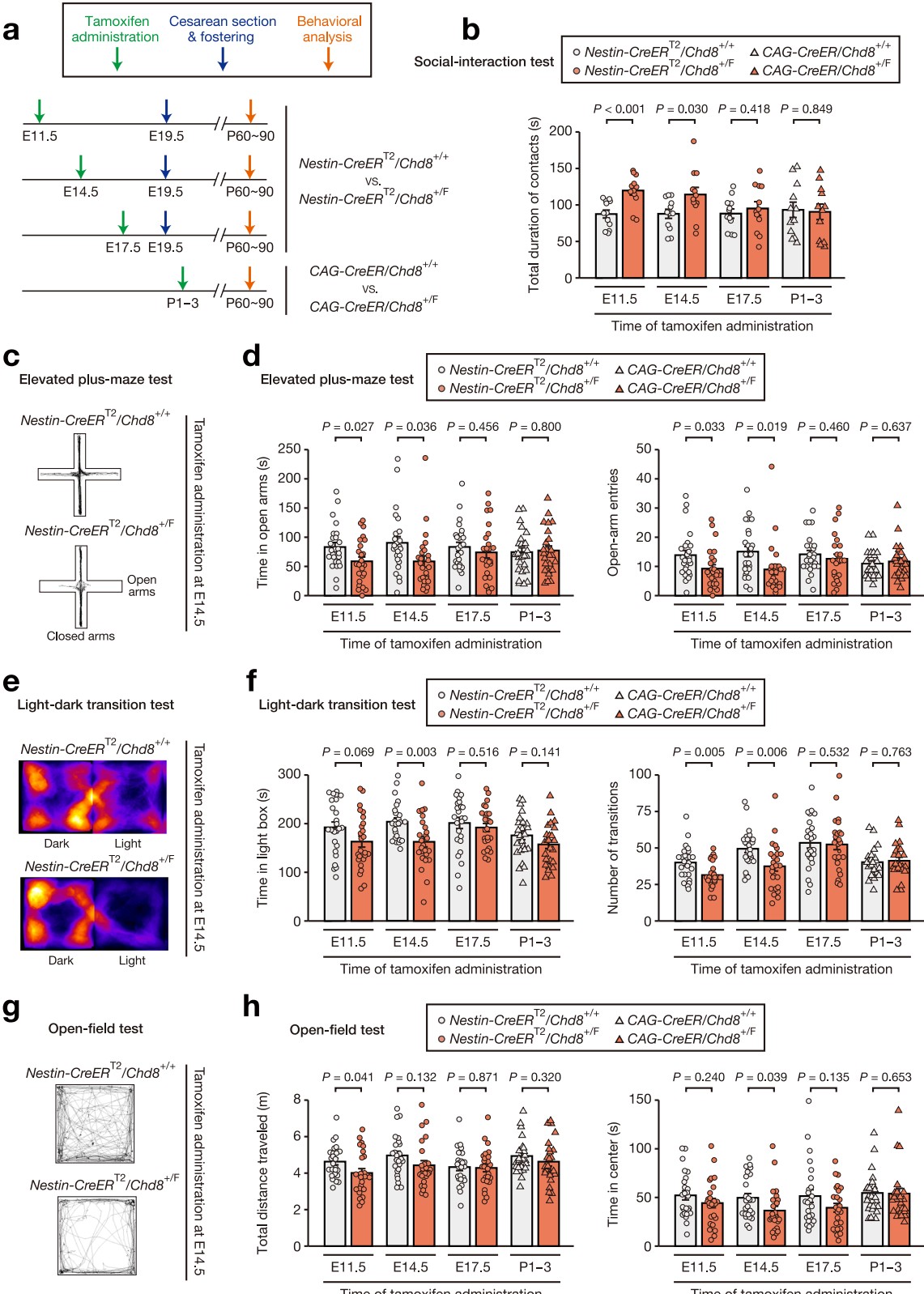

related to inhibitory neurons or to oligodendrocytes were upregulated in the mutant cells of the clusters corresponding to GABAergic neurons and oligodendrocyte-lineage cells, respectively (Fig. 2i and Supplementary Data 2). Focusing on the expression patterns of individual representative genes, we identified numerous differentially expressed genes (DEGs) including *Nrxn1*, *Nlgn1*, and *Neurod1* in excitatory neurons; *Auts2*, *Nrxn3*, and *Gphn* in inhibitory neurons; and *Olig1*, *Tcf4*, and

*Nfia* in OPCs, all of which are related to the differentiation or function of the corresponding cell types and include those associated with autism risk (Fig. 2j and Supplementary Data 1)[40]. Together, these findings suggested that a *Chd8* heterozygous mutation induced during the midfetal period (at E14.5) resulted in substantial changes in gene expression in excitatory neurons, inhibitory neurons, and oligodendrocyte-lineage cells.

**Fig. 1 | Identification of the critical period for the development of autistic-like behavior due to *Chd8* heterozygous mutation. a** Strategy for induction of *Chd8* mutation by tamoxifen administration at the indicated time points, followed by cesarean delivery, fostering, and behavioral assays for *Nestin-CreER*[T2] mice or by behavioral assays for *CAG-CreER* mice. **b** Total duration of contacts for the social-interaction test. **c, d** Representative traces (**c**) as well as time spent in the open arms and number of entries into the open arms (**d**) for the elevated plus-maze test. **e, f** Representative traces (**e**) as well as time spent in the light chamber and number of transitions between the light and dark chambers (**f**) for the light-dark transition test. **g, h** Representative traces (**g**) as well as total distance traveled and time spent

in the central area (**h**) for the open-field test. All quantitative data are means ± s.e.m. and were obtained with *Nestin-CreER*[T2]/*Chd8*[+/+] and *Nestin-CreER*[T2]/*Chd8*[+/F] adult male mice treated with tamoxifen at E11.5, E14.5, or E17.5, or with *CAG-CreER/Chd8*[+/+] and *CAG-CreER/Chd8*[+/F] adult male mice treated with tamoxifen at P1–3. For the social-interaction test only, $n = 12$ pairs per genotype per induction time (one pair = two unfamiliar mice), whereas $n = 25$ mice per genotype per induction time for all other tests. All behavioral tests were conducted with mice at 9 to 13 weeks of age. *P*-values were calculated with the two-tailed Student's *t* test. For (**b**) (E11.5), exact $P = 7.7 \times 10^{-4}$.

## *Chd8* heterozygous mutation is associated with accelerated cell-cycle exit and differentiation in ventral progenitor cells

To assess the impact of *Chd8* mutation on the temporal dynamics of brain development, we next performed trajectory analysis. RNA velocity analysis of our scRNA-seq data revealed dynamic transitions from progenitor cells to various types of neurons and glial cells (Fig. 3a). We then applied the CellRank algorithm to predict the fate probabilities for excitatory neurons, inhibitory neurons, and oligodendrocytes (Fig. 3b). In the case of excitatory neurons, the mature cell population, marked by genes such as *Rbfox1*, was similar for control and mutant groups (Fig. 3c). In contrast, for inhibitory neurons and oligodendrocytes, the mature cell populations—marked by genes such as *Gad1* and *Mbp*, respectively—were significantly larger in the mutant (Fig. 3c).

To analyze further the differences observed in the fate probability analysis, we examined the primary effects of *Chd8* mutation on embryonic neural development. During embryonic neurogenesis, excitatory neurons originate from the ventricular zone of the dorsal telencephalon, whereas inhibitory neurons and oligodendrocyte-lineage cells originate from the ventricular zone of the ventral telencephalon (that is, the ganglionic eminence)[41–45]. Cre-mediated recombination was efficiently induced in SOX2-positive progenitor cells in the dorsal and ventral ventricular zone of *Nestin-CreER*[T2] mice as early as 6 h after tamoxifen administration at E14.5 (Supplementary Fig. 7a–d). We confirmed that the abundance of CHD8 in tdTomato-positive cells with a *Chd8* heterozygous mutation induced at E14.5 was significantly lower than that in those of control mice beginning at E16.5 (Supplementary Fig. 7e–g). On the basis of these findings, we investigated the differentiation status of dorsal and ventral cells at E16.5 by 5-ethynyl-2′-deoxyuridine (EdU) pulse-chase analysis. Pulse-labeling with EdU for 6 h at E16.5 revealed that the *Chd8* heterozygous mutation induced at E14.5 was associated with a significantly higher proportion of Ki67-negative EdU+ cells (cells that had exited the cell cycle) in the ganglionic eminence but not in the cortex (Fig. 3d–f). In addition, EdU pulse-labeling for 24 h between E15.5 and E16.5 confirmed that the frequency of cells in the differentiating layer of the cortex, marked by DCX, was similar between control and mutant mice, whereas the frequency of differentiating interneurons and oligodendrocytes—marked by DLX1 and PDGFRα, respectively—was significantly higher in the ganglionic eminence of the mutant mice (Fig. 3g–m). These results suggested that the midfetal induction of *Chd8* heterozygous mutation results in acceleration of cell-cycle exit and differentiation of ventral progenitor cells.

## Spatial transcriptome analysis reveals that *Chd8* mutation is associated with region-specific changes in gene expression and cell communication in the adult brain

To examine the impact of altered neurogenesis due to *Chd8* mutation on the adult mouse brain, we performed spatial transcriptome analysis with the use of a CosMx Spatial Molecular Imager and 1000-Plex RNA Mouse Neuroscience Panel[46]. The analysis was performed with coronal sections of the cortex and striatum from the brain of control (global *Chd8* wild-type; *Chd8*[+/+]) and mutant (global *Chd8* heterozygous; *Chd8*[+/−]) mice at 8 weeks of age (Fig. 4a). After data integration, we

visualized 14 clusters by UMAP and detected specific expression of marker genes in each cluster (Fig. 4b, c and Supplementary Fig. 8a). These clusters and the expression of the representative genes were spatially visualized in the cortex and striatum, covering distinct regions including cortical layers, white matter, and ventricular areas (Fig. 4d and Supplementary Fig. 8b, c). The cell populations in each cluster were similar between the control and mutant groups (Fig. 4e, f). Comparisons of gene expression between the mutant and control brain across each cluster revealed several DEGs, with both unique and overlapping expression changes in various cell types and regions (Fig. 4g–l, Supplementary Fig. 8d, and Supplementary Data 3). For the layer II–IV neuron cluster, we observed an increase in immediate-early gene expression together with a decrease in synapse-related and GABA receptor gene expression in the mutant, with these expression changes including several layer-specific changes (Fig. 4g, h). For the D1-type medium spiny neuron (D1-MSN) cluster, the expression of *Drd1* as well as of several genes encoding ion channels and neurotransmitter receptors was increased in the mutant (Fig. 4i). About half of the DEGs for the D1-MSN cluster were shared with the D2-MSN cluster (Fig. 4j). For the interneuron cluster, the expression of *Slc32a1*, which encodes vesicular inhibitory amino acid transporter (VGAT), was down-regulated in the mutant cortex (Supplementary Fig. 9a, b and Supplementary Data 3). In addition, oligodendrocytes in the cortex and striatum showed correlated changes in gene expression, with a common decrease in the expression of myelin-related genes such as *Mag*, *Mog*, and *Plp1* in the mutant (Fig. 4k, l).

We then investigated the effect of these gene expression changes in each cluster on intercellular communication networks with the use of the NeuronChat algorithm, which predicts the strength of connections on the basis of the expression of genes for neurotransmitter receptors and those related to signaling pathways[47]. Given the heterogeneity of interneuron subtypes, we further subdivided the interneuron cluster into specific subclusters, distinguishing cortical subtypes such as parvalbumin (PV), somatostatin (SST), and vasoactive intestinal peptide (VIP) neurons as well as striatal subtypes including PV, SST/neuropeptide Y (NPY), and tyrosine hydroxylase (TH) neurons (Supplementary Fig. 8e–g). We found that interneurons manifested downregulation of GABA signaling, resulting in decreased interactions with excitatory neurons across various layers in the cortex of *Chd8* mutant mice (Fig. 4m and Supplementary Fig. 9c). In contrast, intercellular communication among MSNs, choline acetyltransferase (ChAT) neurons, and interneurons was upregulated in the striatum of *Chd8* mutant mice (Fig. 4n and Supplementary Fig. 9d). Together, our spatial transcriptome analysis of the adult brain revealed that CHD8 haploinsufficiency resulted in dysregulation of gene expression and alterations in cell-to-cell communication in inhibitory neurons in a region-dependent manner.

## Functional deficits in excitatory and inhibitory neuronal networks in *Chd8* mutant mice

Having uncovered transcriptomic changes associated with CHD8 haploinsufficiency, we next investigated neuronal function in *Chd8* mutant mice. To assess inhibitory connectivity in vivo, we used fiber photometry combined with optogenetic manipulation in the medial

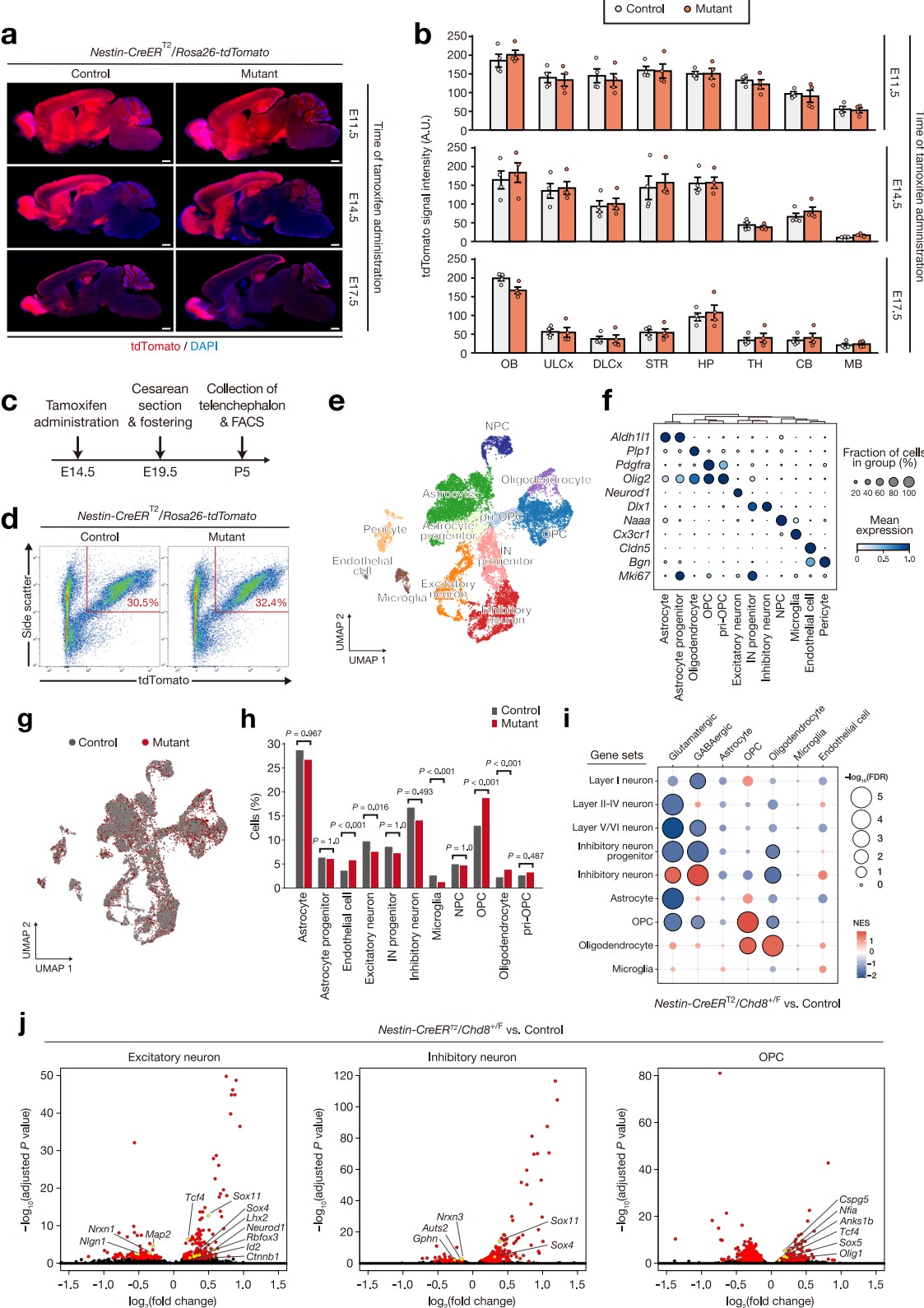

**j** *Nestin-CreER*[T2]/*Chd8*[+/F] vs. Control

prefrontal cortex. We first validated successful optogenetic activation of inhibitory neurons. We injected the adeno-associated virus (AAV) vectors AAV-Syn-FLEX-GCaMP6f and AAV-Syn-FLEX-ChrimsonR-tdTomato[48] into *Vgat-ires-Cre* mice[49], in which Cre recombinase is expressed specifically in GABAergic interneurons that express *Slc32a1* (*Vgat*), and implanted an optic fiber above the injection site (Fig. 5a–c). This strategy achieved Cre-dependent and therefore interneuron-

specific expression of both the Ca²⁺ indicator GCaMP6f and the red-shifted opsin ChrimsonR, allowing simultaneous optical activation and Ca²⁺ signal recording. Light stimulation of ChrimsonR at 648 nm resulted in an immediate increase in Ca²⁺-dependent GCaMP6f fluorescence (Fig. 5d), whereas the Ca²⁺-independent signal remained unchanged (Fig. 5e), confirming successful activation of inhibitory neurons.

**Fig. 2 | Combination of lineage tracing and scRNA-seq analysis reveals cell type–specific changes in gene expression due to midfetal induction of *Chd8* heterozygous mutation. a, b** Representative images of tdTomato immuno-fluorescence staining for sagittal brain sections from 8-week-old *Nestin-CreER*[T2]*/Rosa26-tdTomato/Chd8*[+/+] (control) and *Nestin-CreER*[T2]*/Rosa26-tdTomato/Chd8*[+/F] (mutant) mice and after tamoxifen administration at E11.5, E14.5, or E17.5 (**a**), and quantification of tdTomato signal intensity (data are means ± s.e.m., *n* = 4 mice per genotype) (**b**). Nuclei were stained with 4′6-diamidino-2-phenylindole (DAPI). Scale bars, 1 mm. A.U., arbitrary unit; OB, olfactory bulb; ULCx, upper-layer cortex; DLCx, deep-layer cortex; STR, striatum; HP, hippocampus; TH, thalamus; CB, cerebellum; MB, midbrain. **c** Protocol for isolation of tdTomato-positive cells from the telencephalon of *Nestin-CreER*[T2]*/Rosa26-tdTomato* mice (control or mutant) treated with tamoxifen at E14.5. **d** FACS gating strategy for sorting tdTomato-positive cells from the telencephalon of control and mutant mice after size-gating and exclusion of dead cells (Supplementary Fig. 6b, c). **e** UMAP for 8681 control and 8940 mutant

tdTomato-positive cells isolated from the mouse telencephalon at P5 after tamoxifen treatment at E14.5. NPC, neural progenitor cell; pri-OPC, primitive oligodendrocyte progenitor cell; IN, inhibitory neuron. **f** Dot plot representing the expression of signature genes across the 12 clusters. **g, h** UMAP (**g**) and bar graph (**h**) representing the distribution and percentage of the various cell types among control (gray) and mutant (red) cells. The *P*-values were calculated using Fisher's exact test with Bonferroni correction for multiple comparisons. The exact *P*-values for endothelial cell, microglia, OPC, and oligodendrocyte clusters in (**h**) were $3.1 \times 10^{-10}$, $3.4 \times 10^{-9}$, $3.1 \times 10^{-22}$, and $7.9 \times 10^{-9}$, respectively. **i** GSEA for expression of cell type–specific gene sets in the indicated clusters of mutant mice compared with control mice. FDR, false discovery rate; NES, normalized enrichment score. **j** Volcano plots for DEGs in the indicated clusters of mutant versus control mice. DEGs (FDR-adjusted *P*-value of < 0.05) are shown in red. Genes related to the differentiation or function of each cell type are highlighted in yellow. Wilcoxon rank-sum test (two-sided) with Benjamini–Hochberg correction in (**j**).

We then examined whether CHD8 haploinsufficiency alters the function of inhibitory neuronal circuits. To this end, we induced ChrimsonR expression specifically in inhibitory neurons while expressing GCaMP6f under the control of the Ca²⁺/calmodulin-dependent protein kinase IIα (CaMKIIα) gene promoter in the medial prefrontal cortex (Fig. 5f). Although the CaMKIIα gene promoter has been described as specific for excitatory neurons, the short promoter sequence commonly included in AAV vectors is not strictly specific to excitatory neurons and is active also in interneurons[50]. Consistent with this nonspecific promoter activity, whereas most GCaMP6f-positive neurons were negative for ChrimsonR-tdTomato, a small proportion showed coexpression (Fig. 5g, h).

In control mice, 648-nm light stimulation evoked a biphasic response in the Ca²⁺-dependent signal (Fig. 5i). Immediately after stimulus onset, Ca²⁺-dependent fluorescence showed a rapid, transient increase, likely reflecting direct optogenetic activation of the small population of ChrimsonR/GCaMP6f double-positive inhibitory neurons. This initial increase was followed by a slower late-phase decrease in the Ca²⁺-dependent signal that dropped below the baseline level, consistent with synaptic inhibition exerted by the activated inhibitory neurons on surrounding GCaMP6f-positive neurons. Light stimulation did not elicit a response in the Ca²⁺-independent signal (Fig. 5j).

In *Chd8* mutant mice, the evoked Ca²⁺-dependent signal was higher than that in control mice over the entire course of the biphasic response (Fig. 5i). The time course of the difference in the Ca²⁺ signal between the control and mutant mice did not resemble the early-phase response; instead, it showed a slower profile that overlapped with the late-phase component. This result indicated that the difference was unlikely due to an increased early-phase direct activation of ChrimsonR-expressing neurons in the mutant mice, but more likely reflected a decrease in the late-phase synaptic inhibitory response in the mutant mice. Quantification of the light-evoked Ca²⁺-dependent response within the first 300 ms after stimulation onset revealed a significantly larger signal in *Chd8* mutant mice than in the control animals (Fig. 5k), reflecting the absence of or a reduction in the late-phase undershoot below baseline in the mutant. These findings were thus suggestive of impaired functional connectivity of GABAergic neurons in the medial prefrontal cortex of adult *Chd8* mutant mice.

To characterize further the functional properties of excitatory and inhibitory neurons individually, we performed microelectrode array (MEA) recordings. For the assessment of excitatory neuronal activity, we prepared primary cultures from the E13.5 cortex, which were predominantly excitatory (Supplementary Fig. 10a–c). Excitatory neurons from *Chd8* mutant embryos manifested a significant reduction in mean firing rate, whereas spike amplitude and total axonal length were similar to those for control mice (Supplementary Fig. 10d–h), indicative of hypoactivity in excitatory neurons without an overt change in single-spike amplitude or gross morphology. For investigation of

inhibitory neurons, we established primary cultures from the E13.5 ganglionic eminence (Supplementary Fig. 10a–c)[51]. Inhibitory neurons from *Chd8* mutant embryos showed a significantly shorter axonal branch length compared with control embryos, whereas firing frequency and spike amplitude were unchanged (Supplementary Fig. 10i–m), suggestive of a reduced capacity of inhibitory neurons in the mutant mice to influence surrounding network activity.

Consistent with the impaired inhibitory connectivity for *Chd8* mutant mice indicated by transcriptome analysis (Fig. 4m, n), in vivo functional assay (Fig. 5i, k), and axonal morphology in MEA recordings (Supplementary Fig. 10l, m), immunostaining of the *Chd8* mutant cortex revealed a significant reduction in perisomatic VGAT-positive puncta (Fig. 5l, m). Collectively, these in vivo and in vitro findings were indicative of dysfunction of both excitatory and inhibitory neurons in *Chd8* mutant mice.

## Genetic rescue of *Chd8* expression in neural stem cells at E14.5 or in ventral progenitor cells ameliorates behavioral abnormalities

We next investigated whether the restoration of *Chd8* expression at a specific time or location might prevent the development of phenotypes associated with *Chd8* heterozygous mutation. For this purpose, we generated knock-in mice harboring an LSL cassette preceded by a splicing acceptor site between exons 8 and 9 of the endogenous *Chd8* locus (*Chd8*[+/LSL] mice) (Fig. 6a). *Nestin-Cre/Chd8*[+/LSL] mice manifested CHD8 haploinsufficiency in the testis (absence of Cre-mediated recombination), whereas the expression of CHD8 was restored to near wild-type levels in the brain (with Cre-mediated recombination) (Supplementary Fig. 11a, b). As expected, *Chd8*[+/LSL] mice developed macrocephaly, without a change in body weight (Supplementary Fig. 12a, b), as well as autistic-like behavioral phenotypes including contact abnormality in the reciprocal social-interaction test (Supplementary Fig. 12c, d) and increased anxiety-like behavior in the elevated plus-maze, light-dark transition, and open-field tests (Supplementary Fig. 12e–k), similar to the characteristics of *Chd8*[+/−] mice[14,15]. Neural stem cell–specific restoration of CHD8 expression by the *Nestin-Cre* transgene attenuated the development of macrocephaly and behavioral abnormalities in both male and female *Chd8*[+/LSL] mice (Supplementary Fig. 12b–k). We then investigated whether the timing of the restoration of *Chd8* expression influences this improvement in behavioral phenotypes. The genetic rescue of *Chd8* expression at E14.5 or before, but not at E17.5 or after, ameliorated autistic-like behavior in both male and female mice without affecting macrocephaly (Fig. 6b–d, Supplementary Fig. 13, and Supplementary Fig. 14). These findings suggested that, similar to the relation between *Chd8* heterozygous mutation and behavioral abnormalities, the midfetal restoration of *Chd8* expression is critical for improvement of behavioral phenotypes.

Given that we found that neurogenesis was defective in ventral progenitor cells of *Chd8* mutant mice, we examined whether ventral

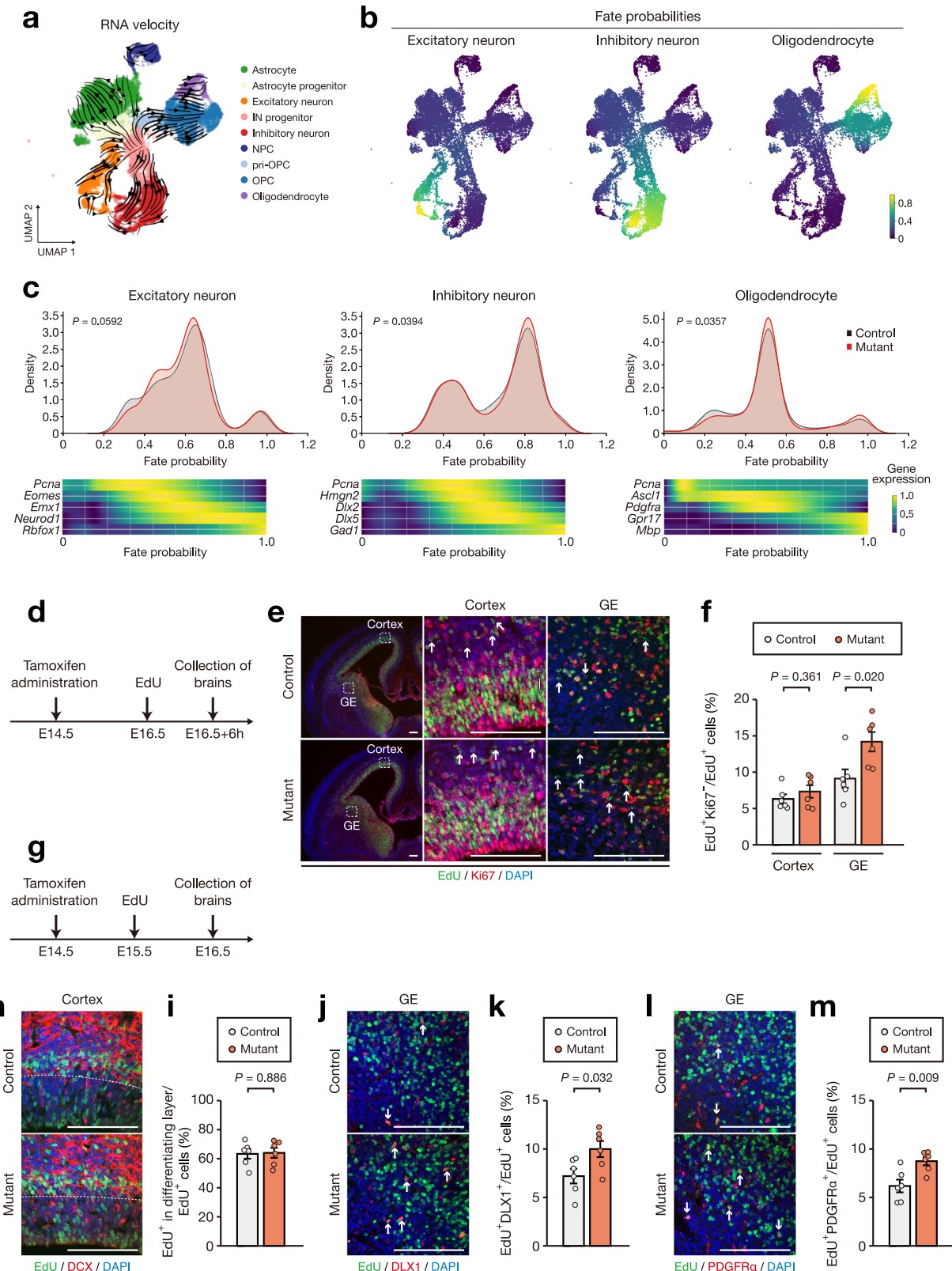

cell–specific restoration of CHD8 expression might also prevent the development of behavioral phenotypes. We previously showed that *Olig1-Cre*–induced mutation of *Chd8* (*Olig1-Cre/Chd8*[+/F] mice) gave rise to autistic-like behavior[26]. *Olig1-Cre* expression was detected in ventral progenitor cells but not in dorsal progenitor cells at E14.5 (Supplementary Fig. 11c–e), consistent with previous results showing that OLIG1 is widely expressed in the progenitors of both oligodendrocytes

and interneurons[52,53]. Behavioral tests showed that autistic-like behavioral phenotypes, but not macrocephaly, were attenuated in both male and female *Olig1-Cre/Chd8*[+/LSL] mice compared with *Chd8*[+/LSL] mice (Fig. 6e–g and Supplementary Fig. 15). We confirmed that restoration of *Chd8* expression in ventral progenitor cells by *Olig1-Cre* resulted in normalization of the differentiation of progenitors of both interneurons and oligodendrocytes (Fig. 6h–j). These results thus indicated

**Fig. 3 | Chd8 heterozygous mutation is associated with accelerated cell-cycle exit and differentiation in ventral progenitor cells. a** RNA velocity field projected onto the UMAP of the scRNA-seq data, with exclusion of the clusters corresponding to microglia, pericytes, and endothelial cells. **b** UMAP visualization highlighting cell fate probabilities directed toward the three terminal states of excitatory neuron, inhibitory neuron, and oligodendrocyte. **c** Density plots (upper panels) for control and mutant cells showing fate probabilities toward excitatory neurons (left), inhibitory neurons (middle), and oligodendrocytes (right), as well as corresponding heat maps (lower panels) representing the expression trends for the indicated genes across the fate probabilities. **d–f** Protocol for EdU pulse-chase analysis of cell-cycle exit (**d**), immunofluorescence staining of Ki67 and EdU (**e**), and quantification of the number of cells negative for Ki67 among EdU[+] cells (**f**) in the cortex and ganglionic eminence (GE) of *Nestin-CreER*[T2]/*Rosa26-tdTomato/Chd8*[+/+] (control) and *Nestin-CreER*[T2]/*Rosa26-tdTomato/Chd8*[+/F] (mutant) embryos at 6 h after pulse-

labeling with EdU at E16.5 and administration of tamoxifen at E14.5 (*n* = 6 mice per genotype). Arrows indicate EdU[+]/Ki67[-] cells. Scale bars, 100 μm. **g–m** Protocol for EdU pulse-chase analysis of progenitor cell differentiation (**g**); immunofluorescence staining of DCX and EdU (**h**) and quantification of the number of cells in the differentiating layer marked by DCX among EdU[+] cells (**i**) in the cortex; immunofluorescence staining of DLX1 and EdU (**j**) and quantification of the number of cells positive for DLX1 among EdU[+] cells (**k**) in the GE; and immunofluorescence staining of PDGFRα and EdU (**l**) and quantification of the number of cells positive for PDGFRα among EdU[+] cells (**m**) in the GE of *Nestin-CreER*[T2]/*Rosa26-tdTomato/Chd8*[+/+] (control) and *Nestin-CreER*[T2]/*Rosa26-tdTomato/Chd8*[+/F] (mutant) embryos at 24 h after pulse-labeling with EdU at E15.5 and administration of tamoxifen at E14.5 (*n* = 6 mice per genotype). Arrows indicate DLX1[+]/EdU[+] and PDGFRα[+]/EdU[+] cells in (**j**) and (**l**), respectively. Scale bars, 100 μm. Data in (**f**), (**i**), (**k**), and (**m**) are means ± s.e.m., and the *P*-values were calculated with the two-tailed Student's *t* test.

that the restoration of *Chd8* expression in ventral progenitor cells is sufficient to ameliorate behavioral phenotypes and the altered differentiation status of ventral cells conferred by CHD8 haploinsufficiency. The generation of *Chd8*[+/LSL] mice to rescue CHD8 expression thus uncovered the critical period and region for correction of behavioral phenotypes.

## Discussion

With the use of behavioral tests and transcriptome analysis, we have here identified in an unbiased manner both the critical period and brain region responsible for the development of behavioral abnormalities due to *Chd8* heterozygous mutation. Among the various alterations in neurogenesis induced by CHD8 haploinsufficiency, our results indicate that accelerated differentiation of ventral progenitor cells is associated with the development of autistic-like behavior. The genetic restoration of *Chd8* expression with an LSL mouse model further corroborated this relation.

*Chd8* heterozygous mutant mice have been studied as a model of ASD because their phenotypes recapitulate key features of humans with this condition due to *CHD8* mutations[14,15]. Similar to other major autism-related proteins, CHD8 is expressed in a variety of cell types and contributes to the developmental processes of multiple cell lineages[22–26]. Given that behavioral abnormalities are central to ASD symptomatology and that their improvement is a primary therapeutic goal[1,3], deciphering the causal relation between neural and behavioral abnormalities is crucial. Our study has now provided unbiased insight into the spatiotemporal identity of specific neural alterations responsible for behavioral abnormalities, with our results implicating ventral progenitor cells, rather than dorsal ones, in behavioral phenotypes.

During embryonic neurogenesis, interneurons and oligodendrocytes originate from the ventricular zone of the ganglionic eminence and migrate to the cortex and striatum[41–45]. Furthermore, intercellular cooperation between interneurons and OPCs is required for their proper differentiation and migration[54,55]. Evidence has suggested that dysregulated differentiation of these ventrally derived cells is associated with the development of ASD[7,56,57]. Several ASD mouse models, including those with *Nlgn3* or *Shank3* mutations, manifest defective differentiation of GABAergic interneurons[7,8,58,59]. On the other hand, proper differentiation of oligodendrocyte-lineage cells and subsequent myelination are important for the development of neural connectivity[60,61]. Mutations in major autism-related genes such as *PTEN* and *MECP2* impair oligodendrocyte differentiation and myelination through effects on the mTOR pathway and NF-κB signaling, respectively[62–65]. Importantly, single-nucleus RNA-seq analysis of postmortem human cortex revealed cell–type-specific and region-dependent changes in gene expression affecting inhibitory neurons and OPCs in individuals with ASD[66], and scRNA-seq analysis of human cerebral organoids showed that *CHD8* disruption compromises the differentiation and thereby increases the production of inhibitory

neurons[23,24]. These human data are consistent with our results obtained with the *Chd8* mouse model, supporting a conserved role for CHD8 in the regulation of inhibitory neurogenesis across species.

Integration of our findings with the embryonic ganglionic eminence, the P5 telencephalon, and adult cortex and striatum suggests a stage-dependent pattern of neurodevelopmental changes. Consistent with a prenatal shift toward differentiation of ventral progenitors induced by *Chd8* mutation as revealed by EdU labeling (Fig. 3d–m), scRNA-seq analysis of the telencephalon at P5 revealed enrichment of differentiation programs across inhibitory neuron, OPC, and oligodendrocyte clusters (Fig. 2i). Of note, whereas OPC and oligodendrocyte fractions were increased in the *Chd8* mutant mice at P5, the interneuron fraction was unchanged (Fig. 2h). This latter result may reflect the physiological wave of early postnatal death of cortical interneurons[67], which might mask prenatal differences in the production of inhibitory neurons. Transcriptome profiling of the adult cortex and striatum revealed downregulation of myelination programs in oligodendrocytes and reduced inhibitory synapse–related gene expression in interneurons of the mutant mice (Fig. 4g–n), indicative of hypomaturation or hypofunction at later stages. Consistent with these molecular signatures, our functional analysis in the medial prefrontal cortex indicated that the inhibitory control exerted by VGAT[+] interneurons over the surrounding network is weakened in adult *Chd8* heterozygous mutant mice. Several ASD models have shown that precocious cell-cycle exit of progenitors leads to transient developmental advancement followed by hypofunction at later stages[27,68]. Indeed, a previous analysis of *Chd8* heterozygous mice found enhanced oligodendrocyte function up to P11 but reduced functionality at adult stages[27], consistent with our current findings. The relations among our findings obtained at different developmental stages require careful interpretation, however. Given that the datasets are cross-sectional snapshots, they should be viewed within a developmental stage–specific framework and do not establish lineage-level causality.

Finally, with the use of newly established *Chd8*[+/LSL] mice, we showed that the development of behavioral abnormalities due to CHD8 haploinsufficiency can be ameliorated by genetic restoration of *Chd8* expression in a spatiotemporally dependent manner. The improvement of behavioral phenotypes by restoration of *Chd8* expression specifically in neural stem cells at E14.5 or in ventral progenitor cells of *Chd8* mutant mice confirmed that the development of autistic-like behavior is dependent on both time and cell type. In addition, the prevention of behavioral phenotypes suggests a degree of plasticity in ASD associated with CHD8 haploinsufficiency and opens avenues for future therapeutic possibilities, as has been shown in other ASD mouse models[69,70]. We have here presented a new strategy to identify the primary relation between neural alterations and behavioral phenotypes, thereby providing a platform that may facilitate the development of treatments for the core symptoms of ASD.

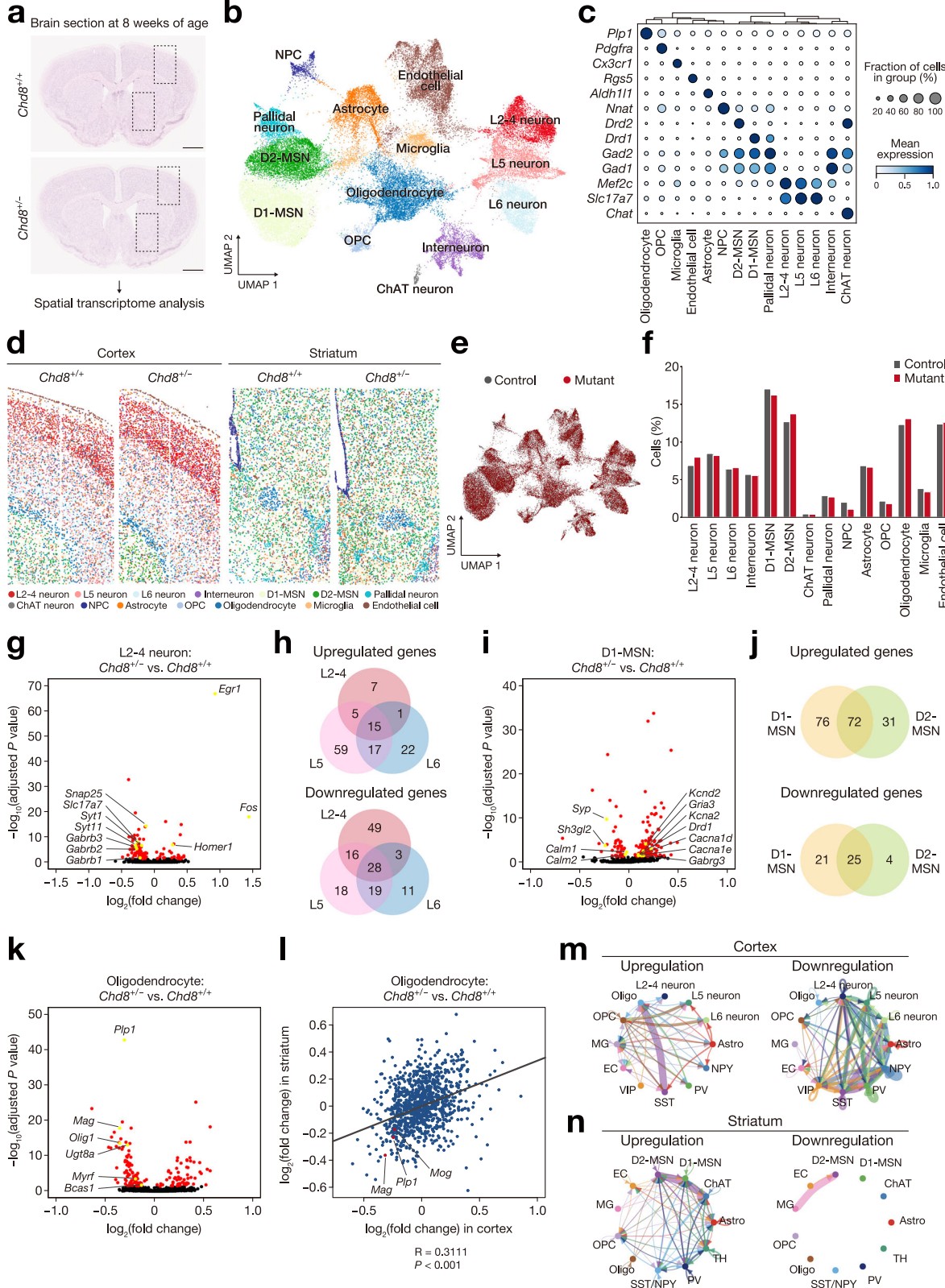

## Methods

### Mice

$Chd8^{+/F}$ mice were generated by flanking exon 11-13 of the $Chd8$ gene with loxP sites, and $Chd8^{+/-}$ mice were obtained by Cre-mediated deletion of the floxed allele[14]. $Chd8^{+/F}$ mice were crossed with $Nestin$-$Cre$, $Nestin$-$CreER^{T2}$, or $CAG$-$CreER$ heterozygous mice to produce $Nestin$-$Cre/Chd8^{+/F}$, $Nestin$-$CreER^{T2}/Chd8^{+/F}$, or $CAG$-$CreER/Chd8^{+/F}$ mice,

respectively. $Rosa26$-$tdTomato$ mice harboring a tdTomato reporter gene with a CAG promoter and LSL cassette at the $Rosa26$ locus were crossed with $Nestin$-$CreER^{T2}$ or $Olig1$-$Cre$ mice to produce $Nestin$-$CreER^{T2}/Rosa$-$tdTomato$ and $Olig1$-$Cre/Rosa$-$tdTomato$ mice, respectively. Offspring of $Chd8^{+/F}$, $Nestin$-$Cre$, $Nestin$-$CreER^{T2}$, $CAG$-$CreER$, $Olig1$-$Cre$, and $Rosa$-$tdTomato$ mice were backcrossed onto the C57BL/6 J line for at least nine generations. $Vgat$-$ires$-$Cre$ mice were generated

**Fig. 4 | Spatial transcriptome analysis reveals that *Chd8* mutation is associated with region-specific changes in gene expression and cell communication in the adult brain. a** In situ hybridization for *Actb* mRNA in serial coronal brain sections from one 8-week-old *Chd8*[+/−] (mutant) and one age-matched *Chd8*[+/+] (control) mouse. Four regions (two cortical and two striatal) per genotype were analyzed. The black boxes represent regions selected for spatial transcriptome analysis. Scale bars, 1 mm. **b** UMAP of 50,505 cells from the cortex and striatum of mutant and control mice. **c** Dot plot showing expression of signature genes across the 14 clusters identified in (**b**). **d** Spatial distribution of each cluster characterized in (**b**). **e, f** UMAP (**e**) and quantification (**f**) of cell-type distribution in control (gray) and mutant (red) cells (two-sided Fisher's exact test with Bonferroni correction). **g** Volcano plot for DEGs in the cortical layer II–IV (L2-4) neuron cluster of mutant versus control. DEGs (FDR-adjusted *P*-value < 0.05) are shown in red. Synapse-, neurotransmitter receptor–, and neuronal activity–related genes are highlighted in yellow. **h** Venn diagrams showing overlap of up- and downregulated DEGs among L2-4, L5, and L6 neuron clusters. **i** Volcano plot for DEGs in the D1-type medium spiny neuron (D1-MSN) cluster of mutant versus control. Synapse- and ion channel–related genes are highlighted in yellow. **j** Venn diagrams showing overlap of up- and downregulated DEGs between D1-MSN and D2-MSN clusters. **k** Volcano plot for DEGs in the oligodendrocyte cluster of mutant versus control. Genes related to myelination are highlighted in yellow. **l** Pearson correlation analysis (two-sided) for fold change in gene expression in the oligodendrocyte clusters of the cortex and striatum for *Chd8* mutant versus control ($P = 6.0 \times 10^{-24}$). **m, n** Circle plots representing upregulated and downregulated intercellular communication networks among cell type clusters in the cortex (**m**) or striatum (**n**) for mutant versus control. Astro, astrocyte; EC, endothelial cell; MG, microglia; Oligo, oligodendrocyte; NPY, PV, SST, VIP, and TH, interneurons positive for neuropeptide Y, parvalbumin, somatostatin, vasoactive intestinal peptide, or tyrosine hydroxylase, respectively. Wilcoxon rank-sum test (two-sided) with Benjamini–Hochberg correction in (**g, i, k**).

by in vitro fertilization with cryopreserved sperm provided by K. Kaneda (Kanazawa University). The original mice were obtained from The Jackson Laboratory (stock *Slc32a1*[tm2(cre)Lowl]/J, strain #:016962), and they had been backcrossed to the C57BL/6 J background for more than five generations before sperm cryopreservation. Mice were genotyped by polymerase chain reaction (PCR) analysis of genomic DNA with primers for *Chd8* (5′-CCCAAAAGACCAAATCAAACAAAC-3′, 5′-CCA TAGGCTGAAGAACCGTAATTG-3′, and 5′-AGGCTTAGAAACCCGTCGA G-3′), *Cre* (5′-AGGTTCGTTCACTCATGGA-3′ and 5′-TCGACCAGTTTAGT TACCC-3′), *Rosa-tdTomato* (5′-CTGTTCCTGTACGGCATGG-3′ and 5′-GGCATTAAAGCAGCGTATCC-3′), and *Vgat-ires-Cre* (5′-CTTCGTCA TCGGCGGCATCTG-3′, 5′-CAGGGCGATGTGGAATAGAAA-3′, and 5′-CCAAAAGACGGCAATATGGT-3′). Sex was not determined for fetal experiments, whereas both male and female mice were used in post-natal experiments, as indicated where applicable. All mice, with the exception of *Vgat-ires-Cre* and *Chd8*[+/−] mice, were maintained under specific pathogen–free conditions, and all experiments were approved by the Animal Care Committee of Kanazawa University (permission no. AP-224334). Experiments with genetically modified animals were approved by the Institutional Biosafety Committee (protocol no. KINDAI 6-2135). Animals were monitored regularly to ensure welfare, and all efforts were made to minimize suffering. Mice were housed under a 12 h light/dark cycle at 23 ± 3 °C and 55 ± 10% humidity, with ad libitum access to food and water. Mice were euthanized by carbon dioxide inhalation in accordance with institutional guidelines.

### Generation of *Chd8* LSL mice
*Chd8* conditional knock-in floxed (*Chd8*[+/LSL]) mice were generated by CRISPR/Cas9-based genome editing in the C57BL/6 J strain. The donor plasmid contained an adenoviral splice acceptor site and a stop cassette flanked by loxP sequences, as described previously[71]. The 5′ and 3′ homology arms of 1075 and 995 bp, respectively, were targeted to the region between exons 8 and 9 of the endogenous *Chd8* locus. The resulting *Chd8* LSL mice were backcrossed onto the C57BL/6 J background for at least five generations to eliminate off-target effects. Mice were genotyped by PCR analysis of genomic DNA with primers for *Chd8*[+/LSL] (5′-CGGGGGTGGGTACACAGACTA-3′, 5′-TAACAACAACGGCG GCTACA-3′, and 5′-GCTAGGCTTCCTAGCTATGCGA-3′).

### Tamoxifen treatment
For induced deletion of *Chd8* in embryos and neonates, pregnant or neonatal mice were treated with tamoxifen as described previously[38], with minor modification. In brief, tamoxifen (Sigma-Aldrich) was dissolved in corn oil (Sigma-Aldrich) to a final concentration of 40 mg/ml for oral administration or of 1 mg/ml for intragastric injection. A single dose of 200 mg/kg was administered by oral gavage to wild-type females made pregnant by *Nestin-CreER*[T2]/*Chd8*[+/F] or *Nestin-CreER*[T2]/*Chd8*[+/LSL] males. Given that tamoxifen administration to pregnant mice interferes with endogenous estrogen-dependent biological processes and can result in difficulty with vaginal delivery, embryos were delivered by cesarean section at E19.5[38] and fostered by C57BL/6 J females. For the induction of Cre-mediated recombination in neonates harboring *CAG-CreER*, the mice were subjected to intragastric injection of 50 µg of tamoxifen at P1, P2, and P3 as described previously[38].

### Antibodies
Antibodies included those to SOX2 (ab97959, abcam; 1:1000 dilution), Ki67 (550609, BD Biosciences; 1:500), DCX (ab18723, abcam; 1:1000), OLIG2 (AB9610, Millipore; 1:500), DLX1 (Af460, Frontier Institute; 1:500), PDGFRα (558774, BD Pharmingen; 1:500), GABA (A2052, Sigma-Aldrich; 1:500), VGAT (14471-1-AP, Proteintech; 1:500), and NeuN (MAB377, Millipore; 1:1000) for immunofluorescence staining, and those to CHD8 (77694, Cell Signaling Technology; 1:1000) and HSP90 (610419, BD Biosciences; 1:1000) for immunoblot analysis. Antibodies to RFP (5f8, ChromoTek; or MA5-15257, Thermo Fisher Scientific, both at 1:1000) were used to enhance the tdTomato signal in immunofluorescence analysis.

### Immunofluorescence staining
Immunofluorescence staining of cryosections was performed as described previously[20,26]. In brief, the mouse brain was fixed with 4% paraformaldehyde in phosphate-buffered saline (PBS) for 4 h at 4 °C. Cryosections (thickness of 16 µm) were exposed to 3% bovine serum albumin and 0.1% Triton X-100 before incubation overnight at 4 °C with primary antibodies. They were then washed with PBS, incubated with Alexa Fluor 488–, Alexa Fluor 555–, or Alexa Fluor 647–conjugated goat secondary antibodies (Thermo Fisher Scientific) for 1 h at room temperature, counterstained with DAPI (Wako), and mounted with the use of Fluoromount (Diagnostic BioSystems). Images were captured with an All-in-One Microscope BZ-X800 (Keyence) or Dragonfly 350 Confocal Microscope System (Andor Technology). BZ-X800 Analyzer or ImageJ software was applied to count the numbers of each cell type. tdTomato signal intensity was measured by quantification of each area after correction for the background signal with ImageJ software.

### Immunoblot analysis
Total protein was extracted from tissue with a lysis buffer (50 mM Tris-HCl (pH 7.5), 150 mM NaCl, 0.5% Triton X-100) supplemented with a protease inhibitor cocktail (Wako). The extracts were subjected to SDS-polyacrylamide gel electrophoresis on an Extra PAGE One Precast Gel (Nacalai Tesque), and the separated proteins were transferred to a 0.2 µm PVDF Mini Format membrane (Bio-Rad) with the use of a Trans-Blot Turbo Transfer System (Bio-Rad). The membrane was exposed to 3% nonfat milk for 30 min before consecutive incubations overnight at

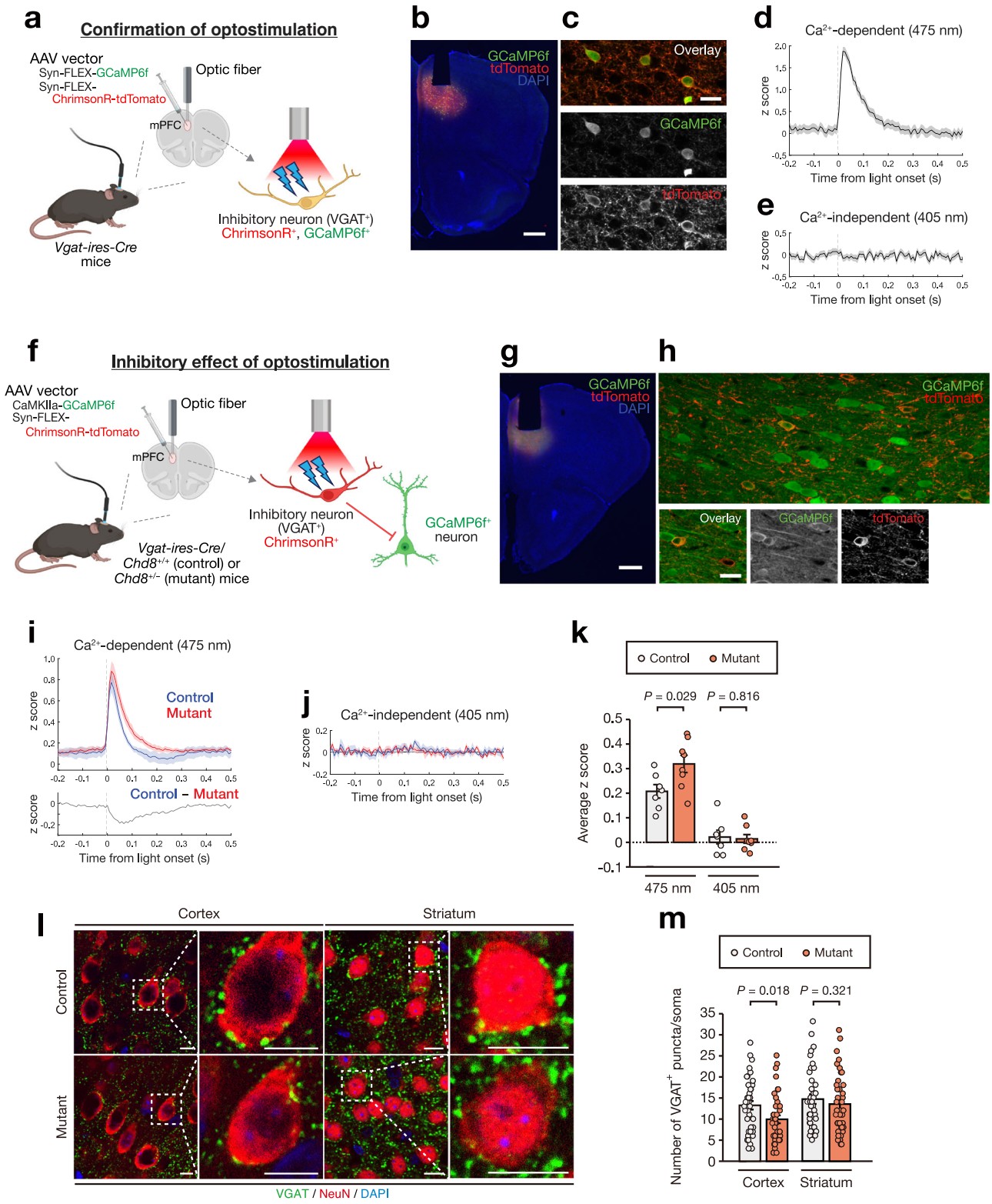

**l** Cortex / Striatum — Control / Mutant

VGAT / NeuN / DAPI

4 °C with primary antibodies and for 1 h at room temperature with horseradish peroxidase-conjugated goat anti-rabbit or anti-mouse secondary antibodies (Thermo Fisher Scientific; 1:20000). Immune complexes were detected with SuperSignal West Pico PLUS Chemiluminescent Substrate (Thermo Fisher Scientific), and images were captured with an ImageQuant LAS 4000 system (GE Healthcare). ImageJ software was applied to measure the signal intensity for each protein.

**EdU labeling**

For analysis of cell-cycle exit, pregnant dams were injected intraperitoneally with EdU (100 mg/kg) at E16.5, 6 h before isolation of embryos for analysis. For analysis of differentiation, pregnant dams were injected intraperitoneally with EdU (100 mg/kg) at E15.5, 24 h before isolation of embryos. The brain was collected from embryos and fixed for 4 h at 4 °C with 4% paraformaldehyde in PBS. Cryosections (thickness of 16 μm) were prepared and incubated for 30 min at

**Fig. 5 | Functional deficits of excitatory and inhibitory neuronal networks in** *Chd8* **mutant mice. a** Schematic representation of the experimental approach to confirm optogenetic stimulation of VGAT⁺ inhibitory neurons. AAV-mediated Cre-dependent coexpression of the Ca²⁺ indicator GCaMP6f and the red-shifted opsin ChrimsonR allows detection of light-induced neuronal activation by fiber photometry. **b** Fluorescence image showing the site of optic fiber implantation and expression of GCaMP6f and (ChrimsonR-)tdTomato in the medial prefrontal cortex. **c** Fluorescence images showing coexpression of GCaMP6f and (ChrimsonR-)tdTomato. **d, e** Stimulus-triggered average traces of z-scored GCaMP6f fluorescence in *Vgat-ires-Cre/Chd8*⁺/⁺ mice. A rapid increase in Ca²⁺-dependent fluorescence (**d**), without an increase in Ca²⁺-independent fluorescence (**e**), indicates successful optogenetic activation of ChrimsonR-expressing interneurons. **f** Schematic representation of the experimental approach to examine the effect of optogenetically activated VGAT⁺ inhibitory neurons on surrounding neurons. AAV-mediated expression of ChrimsonR in VGAT⁺ neurons and GCaMP6f in surrounding neurons allows detection of synaptic inhibition. **g** Fluorescence image showing the site of optic fiber implantation and expression of GCaMP6f and (ChrimsonR-)

tdTomato in the medial prefrontal cortex. **h** Fluorescence images showing that most GCaMP6f-positive neurons were negative for (ChrimsonR-)tdTomato, with a small subset showing coexpression. **i,j** Stimulus-triggered average z-scored traces of GCaMP6f fluorescence in control (*Chd8*⁺/⁺) and mutant (*Chd8*⁺/⁻) mice. Ca²⁺-dependent and Ca²⁺-independent traces are shown in (**i**, upper) and (**j**), respectively. The subtracted trace (control − mutant) is shown in (**i**, lower trace). **k** Mean z scores over the first 300 ms after light stimulation for control (*n* = 7 mice) and mutant (*n* = 8 mice). **l, m** Immunofluorescence staining of VGAT and NeuN (**l**) and quantification of VGAT⁺ puncta per NeuN⁺ soma (**m**) in the cortex and striatum of *Chd8*⁺/⁻ (mutant) and *Chd8*⁺/⁺ (control) mice at 8 weeks of age (*n* = 40 somas per genotype). Scale bars, 500 μm (**b,g**), 20 μm (**c,h**), or 10 μm (**l**). Lines and shaded regions in (**d**), (**e**), (**i**), and (**j**) indicate means ± s.e.m. Data in (**k**) and (**m**) are means ± s.e.m., and *P*-values were calculated with two-tailed Student's *t* test. Parts of Fig. 5a, f were created in BioRender. Created in BioRender. Tashiro, A. (2026) https://BioRender.com/zbnnto5. Created in BioRender. Tashiro, A. (2026) https://BioRender.com/qnnrmmm.

room temperature with PBS containing 0.5 mM CuSO₄, 50 mM ascorbic acid, and 0.5 μM of an azide conjugate of tetramethylrhodamine (5-TAMRA-azide) or fluorescein (6-FAM-azide) before exposure to primary antibodies for immunofluorescence analysis.

## FACS

The telencephalon including the cortex and striatum (without the olfactory bulb and thalamus) was collected from P5 or embryonic mice harboring *Nestin-CreER*ᵀ²/*Rosa26-tdTomato* after tamoxifen treatment. The tissue was dissociated with a papain dissociation system (LK003150, Worthington Biochemical), and the dissociated cells were suspended in PBS containing 2 mM EDTA and 0.5% bovine serum albumin before exposure for 30 min on ice to LIVE/DEAD Fixable Violet Dead Cell Stain reagent (Thermo Fisher Scientific) at a 1:1000 dilution. FACS was performed with a FACSAria Fusion Cell Sorter (BD Biosciences) fitted with a 100-μm nozzle. After gating for forward and side scatter to exclude doublets and debris, dead cells labeled with the LIVE/DEAD dye were removed by gating for Pacific Blue fluorescence. tdTomato-positive cells were then isolated from the P5 or embryonic brain by gating for phycoerythrin fluorescence and were subjected to scRNA-seq or immunoblot analysis, respectively. The data generated during cell sorting were analyzed with FlowJo_V10 software.

## scRNA-seq analysis

For scRNA-seq analysis, duplicate samples of tdTomato-positive cells were isolated by FACS as described above. Each sample of *Nestin-CreER*ᵀ²/*Rosa26-tdTomato/Chd8*⁺/⁺ (control) or *Nestin-CreER*ᵀ²/*Rosa26-tdTomato/Chd8*⁺/ᶠ (mutant) cells was collected from three mice at P5. Libraries were prepared with a Chromium Next GEM Single Cell 3′ Kit v3.1 (10x Genomics) with the input of 5000 cells. The scRNA-seq data were generated with the 10x Genomics Chromium platform. Reads were mapped to the mouse (mm10) genome with the use of Cell Ranger (v7.0.0). Data processing and visualization of scRNA-seq data were performed mostly with Scanpy (v1.9.3)[72]. Cells with < 2000 or > 40,000 total counts were removed, as were those with < 1000 detected genes. In addition, cells with > 10% mitochondrial gene expression were excluded to minimize potential contamination by dead or stressed cells. A total of 8681 control cells and 8940 mutant cells remained for analysis. The average total counts per cell was 12,776.55 for control cells and 12,676.14 for mutant cells. For dimensionality reduction and clustering, principal component analysis was performed with the 2000 highly variable genes selected by the Scanpy function (highly_variable_genes). Batch effects were corrected for by application of the single-cell Variational Inference (scVI) model[73]. Leiden clustering was performed with the neighborhood graph constructed from the scVI latent representation. UMAP was then applied to visualize the batch-corrected latent space. Marker genes

characteristic of specific cell types were used to identify and label clusters. Gene expression was compared between control and mutant cells with the Wilcoxon rank-sum test (scanpy.tl.rank_genes_groups, method = 'wilcoxon' and scanpy.get.rank_genes_groups_df). Gene Ontology analysis of DEGs (FDR *q*-value of < 0.05) was performed with the use of DAVID[74]. GSEA was performed as described previously[75] with the use of GSEA software version 4.3.2. Gene sets specifically expressed in each cell type were defined from RNA-seq or scRNA-seq data of previous studies[76,77].

## RNA velocity and fate probability analysis

Splicing information was extracted from the BAM files generated by Cell Ranger with the use of the velocyto.py tool[78]. scVelo (v0.2.3) was applied to estimate RNA velocity by leveraging splicing kinetics to infer the future state of individual cells on the basis of the ratio of spliced to unspliced transcripts[79]. Cell fate probabilities were estimated with CellRank (v2.0.4)[80] by integration of RNA velocity data with a Markov chain model. RNA velocity and transcriptomic similarity information were combined into a single kernel by weighting PseudotimeKernel and ConnectivityKernel to compute a cell-cell transition matrix. This matrix was then used to estimate cell fate trajectories, and the Generalized Perron Cluster Cluster Analysis (GPCCA) estimator was applied to identify macrostates, representing initial, intermediate, and terminal cell states. Fate probabilities were computed for the terminal states of each cell type identified with GPCCA, including oligodendrocyte, excitatory neuron, inhibitory neuron, NPC, OPC, and astrocyte. Heat maps and density plots were generated to visualize the relation between fate probabilities and cell populations, as well as the expression of cell type–specific genes.

## Sample preparation and data acquisition for spatial transcriptome analysis

Male global *Chd8* wild-type (*Chd8*⁺/⁺) and *Chd8* heterozygous mutant (*Chd8*⁺/⁻) mice at 8 weeks of age were anesthetized by intraperitoneal injection of a mixture of dexmedetomidine hydrochloride (0.75 mg/kg, Wako), midazolam (4.0 mg/kg, Wako), and butorphanol tartrate (5.0 mg/kg, Wako) and then subjected to perfusion with 10% neutral-buffered formalin. The brain was removed from the skull, sliced at a thickness of 5 mm, and exposed overnight to 10% neutral-buffered formalin at room temperature. The tissue was then dehydrated by exposure to an ethanol gradient and embedded in paraffin. Coronal sections (thickness of 5 μm) were prepared from the region 1 mm anterior to the bregma with a microtome at Genostaff (Tokyo, Japan). One section each from control and mutant mice was placed on the same VWR Premium Superfrost Plus microscope slide and analyzed with the CosMx Spatial Molecular Imager (Nanostring)[46] and a 1000-Plex RNA Mouse Neuroscience Panel by Visualix (Kobe, Japan). Fields

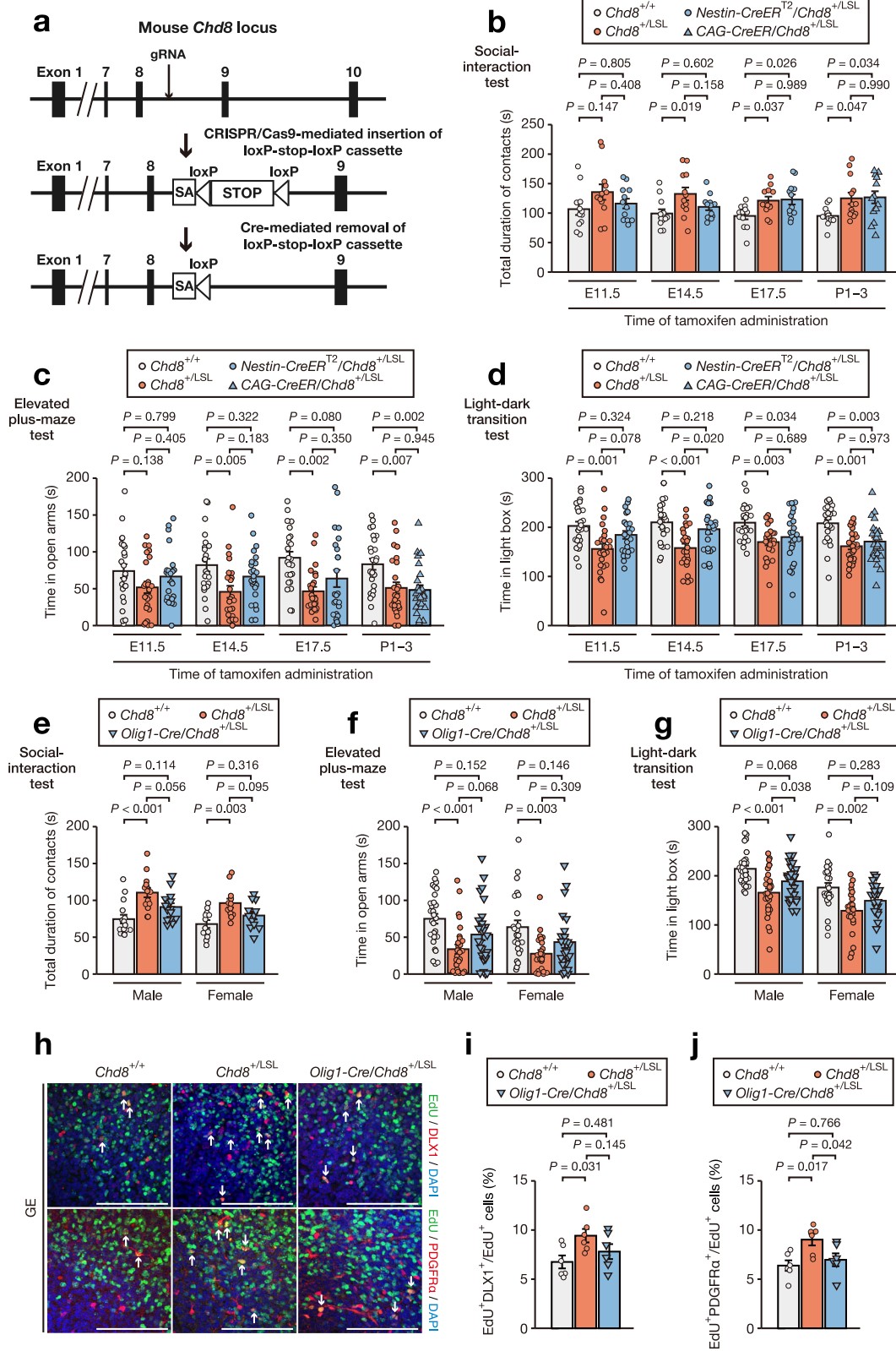

of view (FOVs) measuring 0.51 by 0.51 mm were set to include both the cortex and striatum.

## Data processing for spatial transcriptome analysis

Spatial transcriptomic data were processed with the use of the Scanpy (v1.9.3) and Squidpy (v1.2.4) Python packages. Raw count matrices, metadata, and FOV position files were imported with the

sq.read.nanostring() function. Cells with < 200 total counts were removed, and only cells with at least 120 detected genes were retained. After filtering, 50,505 cells remained for downstream analysis. For dimensionality reduction and clustering, principal component analysis was performed with the 300 highly variable genes selected by the Scanpy function (highly_variable_genes). The scVI model was applied to correct for variation among samples. Leiden clustering was

**Fig. 6 | Genetic rescue of *Chd8* expression in neural stem cells at E14.5 or in ventral progenitor cells ameliorates behavioral abnormalities. a** Schematic depicting the generation of *Chd8*[+/LSL] mice for Cre-dependent rescue of *Chd8* expression. gRNA, guide RNA; SA, splice acceptor. **b–d** Total duration of contacts for the social-interaction test (**b**), time spent in the open arms for the elevated plus-maze test (**c**), and time spent in the light chamber for the light-dark transition test (**d**) in *Chd8*[+/+], *Chd8*[+/LSL], and *Nestin-CreER*[T2]/*Chd8*[+/LSL] adult male mice treated with tamoxifen at E11.5, E14.5, or E17.5 (*n* = 24 per genotype), and with *Chd8*[+/+], *Chd8*[+/LSL], and *CAG-CreER*/*Chd8*[+/LSL] adult male mice treated with tamoxifen at P1–3 (*n* = 26 per genotype). **e–g** Total duration of contacts for the social-interaction test (**e**), time spent in the open arms for the elevated plus-maze test (**f**), and time spent in the light chamber for the light-dark transition test (**g**) in *Chd8*[+/+], *Chd8*[+/LSL], and *Olig1-*

*Cre*/*Chd8*[+/LSL] adult male mice (*n* = 30, 29, and 30 per genotype, respectively) and female mice (*n* = 24 per genotype). All behavioral tests in (**b**) to (**g**) were conducted with mice at 9 to 13 weeks of age. **h–j** Immunofluorescence staining of DLX1 and EdU (upper) and of PDGFRα and EdU (bottom) (**h**) as well as quantification of the number of cells positive for DLX1 among EdU[+] cells (**i**) and of those positive for PDGFRα among EdU[+] cells (**j**) in the ganglionic eminence (GE) of *Chd8*[+/+], *Chd8*[+/LSL], and *Olig1-Cre*/*Chd8*[+/LSL] mice at 24 h after pulse-labeling with EdU at E15.5 (*n* = 6 per genotype). Arrows indicate DLX1[+]/EdU[+] and PDGFRα[+]/EdU[+] cells in (**h**, upper) and (**h**, bottom), respectively. Scale bars, 100 μm. All data are means ± s.e.m. *P*-values were calculated by one-way ANOVA with Tukey's post hoc test. The exact *P*-values for (**d**) (*Chd8*[+/+] versus *Chd8*[+/LSL] at E14.5), (**e–g**) (male *Chd8*[+/+] vs *Chd8*[+/LSL]) were $9.8 \times 10^{-5}$, $2.3 \times 10^{-4}$, $2.5 \times 10^{-4}$, and $2.6 \times 10^{-5}$, respectively.

---

performed with the neighborhood graph constructed from the scVI latent representation. UMAP was then applied to visualize the batch-corrected latent space. Clusters were identified and annotated on the basis of marker genes characteristic of specific cell types. Gene expression was compared between control and mutant cells by the Wilcoxon rank-sum test with the scanpy.tl.rank_genes_groups function (method = 'wilcoxon') and scanpy.get.rank_genes_groups_df. Cell-cell communication analysis was conducted with the NeuronChat R package[47].

## Primary neuronal culture

Neurons were isolated from the cortex and ganglionic eminence of E13.5 mice in order to establish primary cultures of excitatory and inhibitory neurons, respectively[51]. The cells were dissociated by mechanical trituration and plated at a density of $1.5 \times 10^5$ cells per MEA chip or slide coated with poly-D-lysine (P6407, Sigma-Aldrich) for MEA recording or immunostaining, respectively. The cultures were maintained at 37 °C for 48 h in neurobasal medium supplemented with 2% B27 (17505055, Thermo Fisher Scientific), 2 mM L-glutamine (25030081, Thermo Fisher Scientific), and 1% penicillin-streptomycin (15140122, Thermo Fisher Scientific). Cytosine arabinoside (1 μM) was then added for 24 h to eliminate proliferating cells. The cells were subsequently maintained for an additional 21 days before MEA recording or immunostaining.

## MEA recording

Neuronal activity was recorded with the MaxOne high-density microelectrode array system (MaxWell Biosystems, Zurich, Switzerland)[81]. Each chip contains 26,400 electrodes over an area of $3.85 \times 2.10$ mm$^2$ (3265 electrodes/mm$^2$) and with an interelectrode separation of 17.5 μm. Data were acquired at a sampling rate of 20 kHz and analyzed with MaxLab Live software (version 25.2.0). For localization of active units, the ActivityScan Assay was performed with a Sparse 7x configuration scan; each configuration was sequentially recorded for 30 s. For analysis of the activity of each neuron unit, the AxonTracking Assay was performed with the positioning determined by the ActivityScan. Unit selection prioritized spike amplitude, and the minimum spacing between units was set to 175 μm. The outputs of AxonTracking were then processed through Spike Sorting and Footprint Extraction to separate each neuronal activity unit with a minimum spike count threshold of 20 and a footprint completeness threshold of 0.75. Up to 30 units were recorded per well, and we quantified mean firing rate, axonal length, and spike amplitude at the initiation site as well as conduction velocity for the top 10 units ranked by firing rate (or all units when fewer than 10 were available).

## General protocol for behavioral tests

Mice of the indicated genotypes were group-housed (three to four animals per cage) in a room with a 12-h-light, 12-h-dark cycle (lights on at 8:00 a.m.) and with free access to food and water. Behavioral tests were performed with male or female mice at 9 to 13 weeks of age

between 9:00 a.m and 7:00 p.m. as described previously[14,26,82]. Each apparatus was cleaned with a dilute sodium hypochlorite solution before each testing to prevent bias due to olfactory cues. Behavioral tests were conducted as a battery with the same cohort of mice in the following order: open-field test, light-dark transition test, elevated plus-maze test, sociability and social-novelty tests, social-interaction test, and self-grooming test. The hot-plate test and cognitive flexibility test were performed sequentially with another cohort of mice. Each test was separated by an interval of at least 24 h. The social-interaction test, sociability and social-novelty preference tests, light-dark transition test, elevated plus-maze test, and open-field test were conducted with automated analysis systems as described below, whereas the self-grooming test, attentional set-shifting task, and hot-plate test were manually assessed by well-trained experimenters through observation of captured videos. The experimenters were always blinded to mouse genotype in order to exclude bias in behavioral measurements. Data acquisition for mouse locomotion was performed with a camera at a rate of three frames per second.

## Social-interaction test

The social-interaction test was conducted to measure social behavior in a novel environment, as described previously[82]. Two mice of the same sex and genotype that had previously been housed in different cages were placed into the opposite corners of an acrylic box (40 by 40 by 30 cm) and allowed to explore freely for 10 min. The total number of contacts, total duration of contacts, and total distance traveled were calculated automatically with in-house software written in Python.

## Sociability and social-novelty preference tests

The three-chamber social-approach test is adopted to investigate sociability and preference for social novelty[83]. The testing apparatus consisted of a rectangular three-chambered box (O'Hara & Co., Tokyo, Japan). Each chamber was 20 by 40 by 30 cm, and the dividing walls were made of clear Plexiglas and contained a small square opening (6 by 6 cm) that allowed access into each chamber. The tests were performed as previously described[82]. In brief, a subject mouse was placed in the box and allowed to explore for 10 min before the test (habituation session). During the following session, an unfamiliar C57BL/6 J mouse of the same sex (stranger 1) was placed into a wire cage located in one of the side chambers, and an identical empty wire cage was placed in the opposite side chamber to provide a nonsocial stimulus. The right-left positions of the stranger-containing cage and the empty cage were systematically alternated between trials. The subject mouse was then placed in the central compartment and allowed to explore the entire box for 10 min for assessment of sociability (sociability test). Next, a second stranger mouse of the same sex was placed into the wire cage in the other outside compartment that had been empty during the first 10-min test session, so as to evaluate social preference for a new stranger over 10 min (social-novelty preference test). The subject mouse thus had a choice between the first, already-investigated, and now-familiar mouse (stranger 1) and the novel, unfamiliar mouse

(stranger 2). The amount of time spent in each chamber during each session was analyzed automatically with in-house software written in Python. The sociability index was calculated as the ratio of the difference in the time spent in the chamber with stranger 1 and that spent in the chamber with the empty cage to the sum of the time spent in each side chamber in the sociability test[83]. The social-novelty index was calculated as the ratio of the difference in the time spent in the chamber with stranger 2 (novel mouse) and that spent in the chamber with stranger 1 (familiar mouse) to the sum of the time spent in each side chamber in the social-novelty preference test[83].

### Light-dark transition test

The light-dark transition test is widely adopted to measure anxiety-like behavior in mice and was performed as previously described[84]. The apparatus comprised a cage (21 by 42 by 25 cm) divided into two sections of equal size by a partition with a door (O'Hara & Co.). In each test, a subject mouse was placed into the dark chamber with the door closed and was then allowed to move freely between the two chambers for 10 min with the door open. The distance traveled, total number of transitions between the two compartments, latency to first entry into the light chamber, and time spent in the light chamber were recorded automatically with in-house software written in Python.

### Elevated plus-maze test

The elevated plus-maze test, also widely used to assess anxiety-like behavior, was performed as previously described[85]. The apparatus consisted of two arms without walls (open arms, 25 by 5 cm), two arms of the same size including transparent walls with a height of 10 cm (closed arms), and a central square (5 by 5 cm) connecting the arms, which were at an angle of 90 degrees to each other (O'Hara & Co.). The arms and central square were made of white plastic plates and were elevated to a height of 55 cm above the floor. The open arms were surrounded by a raised ledge (3 mm in thickness and height) to prevent mice from falling off the apparatus. Arms of the same type were located opposite one another. Each mouse was placed in the central square at the beginning of each test, and the number of arm entries, distance traveled, and time spent in the open arms over 10 min were recorded automatically with in-house software written in Python.

### Open-field test

Locomotor activity was measured with the open-field test. A subject mouse was placed in the corner of an open-field apparatus (50 by 50 by 40 cm, O'Hara & Co.), which was illuminated at 100 lux. The total distance traveled and time spent in the central area (25 by 25 cm) were recorded over 10 min and analyzed automatically with in-house software written in Python. In addition, perimeter circling was quantified during the same session to assess motor stereotypies. A rotation was scored when a subject mouse progressed unidirectionally along the periphery (within 5 cm of the outer wall)[86].

### Self-grooming test

Repetitive behavior was evaluated with the self-grooming test, as previously described[14]. Each mouse was placed individually into a new transparent cage (10 by 25 by 15 cm). After a 10 min habituation period, the activity of the mouse was videotaped, and the time spent engaged in grooming behavior (paw licking, body grooming or scratching, and head, hind leg, or genital washing) was calculated manually by the experimenters.

### Attentional set-shifting task

Cognitive flexibility was evaluated with the attentional set-shifting task (ASST), as previously described[87,88]. A U-maze apparatus comprised a starting compartment (11 by 19 cm) opening onto two identical reward compartments (each 21 by 9.5 cm). Mice were subjected to food restriction (1 g of chow/day) for 4 days before and throughout testing

so to maintain ~85% of their free-feeding body weight, with water being available ad libitum. On day 1, the subject mouse was habituated to the chamber and digging bowls. On day 2, testing comprised simple discrimination (SD), compound discrimination (CD), and reversal (REV). On day 3, mice completed two intradimensional shifts (IDS1, IDS2) and the extradimensional shift (EDS). In each trial, a buried reward was paired with odor or digging-medium cues, with side and cue identities being counterbalanced. In brief, SD required discrimination within one relevant dimension; CD included a non-predictive second dimension; REV involved inverted contingencies; IDS used novel exemplars within the same relevant dimension; and EDS required shifting to the previously irrelevant dimension. A choice was defined as digging. The criterion was six consecutive correct choices (maximum of 30 trials per stage), and mice not achieving the criterion were excluded.

### Hot-plate test

Thermal nociception was assessed with the hot-plate test. Mice were placed on a heated surface at a temperature of 55 °C, and the latency to the first nocifensive response (hind-paw lick, foot shake, or jump) was recorded with a cutoff time of 15 s.

### Viral vectors

Virus preparations were obtained from Addgene and included pENN.AAV.CamKII.GCaMP6f.WPRE.SV40 (provided by James M. Wilson, Addgene viral prep # 100834-AAV1; http://n2t.net/addgene:100834; RRID:Addgene_100834), pAAV.Syn.Flex.GCaMP6f.WPRE.SV40[89] (provided by Douglas Kim and GENIE Project, Addgene viral prep # 100833-AAV1; http://n2t.net/addgene:100833; RRID:Addgene_100833), and pAAV-Syn-FLEX-rc[ChrimsonR-tdTomato][48] (provided by Edward Boyden, Addgene viral prep # 62723-AAV1; http://n2t.net/addgene:62723; RRID:Addgene_62723).

### Viral vector injection and optic fiber implantation

Fiber photometry was performed with *Vgat-ires-Cre*$^{+/-}$/*Chd8*$^{+/+}$ and *Vgat-ires-Cre*$^{+/-}$/*Chd8*$^{+/-}$ mice. A mixture of AAV-Syn-FLEX-GCaMP6f and AAV-Syn-FLEX-ChrimsonR-tdTomato was diluted in PBS to a final titer of $2.0 \times 10^{12}$ genome copies/ml for each vector. Mice were anesthetized with isoflurane and injected with 0.2 µl of the virus mixture at stereotaxic coordinates of 2.43 mm anterior and 0.45 mm lateral (right) to the bregma, and 1.20 mm ventral to the brain surface. An optic fiber cannula was implanted at the same anterior-posterior and medial-lateral coordinates, with the fiber tip positioned 0.95 mm ventral to the brain surface, during the same surgery.

### Fiber photometry

Recording was performed 21 to 47 days after viral vector injection and fiber cannula implantation and with the use of a fiber photometry system (Doric Lenses, Quebec, Canada) connected to a Fluorescence Mini Cube for monitoring of GCaMP6f fluorescence (Doric Lenses) that was equipped with a 400–410 nm excitation filter (Ca$^{2+}$-independent signal), 460–490 nm excitation filter (Ca$^{2+}$-dependent signal), and 500- to 540-nm emission filter. Excitation light at 405 nm and 475 nm was generated by light-emitting diodes (CLED_405 and CLED_HP_475, Doric Lenses), and 648-nm light for ChrimsonR excitation was generated with an OBIS FP 1193843 laser (Coherent). All excitation sources were combined in the Fluorescence Mini Cube for delivery through a single optic fiber, allowing simultaneous monitoring of GCaMP6f fluorescence and ChrimsonR activation. The fiber cable was connected to a rotary joint (Doric Lenses) and thence to another optic fiber cable leading to the fiber cannula implanted in the subject mouse. For ChrimsonR stimulation, transistor-transistor logic (TTL) pulses (0.5-ms in duration) were delivered, with a power of 2.8 to 3.0 mW measured at the tip of the fiber cable before connection to the fiber cannula. Up to 90 light stimulation trials were performed per recording session, with

~ 10-s intervals (total duration up to ~ 15 min). For recordings, cable-connected mice were placed individually in a cage identical in shape and material to the home cage. Data obtained from a total of 180 to 480 trials per animal are presented. Fluorescence signals were recorded at 120 Hz. The arPLS baseline-estimation algorithm[90] (https://github.com/heal-research/arPLS) was applied for the extraction of slowly varying components from the fluorescence traces obtained with 475 nm and 405 nm excitation. The resulting baseline was subtracted from the raw signal, and the baseline-corrected fluorescence traces were used for all subsequent analyses.

### Histological confirmation of virus transduction and recording location

After completion of fiber photometry recordings, mice were subjected to intracardiac perfusion with 4% paraformaldehyde in 0.1 M phosphate buffer, and the head was then fixed in the same solution with the implanted cannula in place for at least 1 day. Coronal brain sections (thickness of 40 µm) were prepared to verify that the ventral tip of the optic fiber was located in the medial prefrontal cortex and that GCaMP6f and ChrimsonR-tdTomato expression was detected below the ventral tip of the fiber. All examined mice (n = 17) met these criteria, and so none was excluded from the analysis.

### Statistical analysis

Quantitative data are presented as means ± s.e.m. unless indicated otherwise, with the number of mice subjected to each experiment also being stated. Statistical analysis by the unpaired Student's *t* test or by one-way analysis of variance (ANOVA) with Tukey's post hoc test was performed with the use of the R language. A *P*-value of < 0.05 was considered statistically significant.

### Reporting summary

Further information on research design is available in the Nature Portfolio Reporting Summary linked to this article.

## Data availability

The scRNA-seq data have been deposited in GEO under the accession number GSE278323 and in DDBJ Sequence Read Archive under the accession number DRA016611 [https://ddbj.nig.ac.jp/search/entry/bioproject/PRJDB14499]. Spatial transcriptome analysis data have been deposited in GEO under the accession number GSE278133 [https://www.ncbi.nlm.nih.gov/geo/query/acc.cgi]. All other data are available within the Source Data file. Source data are provided in this paper.

## Code availability

The codes for the scRNA-seq and spatial transcriptome analyses used in this study can be found on GitHub: https://github.com/akwamura/Chd8_scRNA-seq, and archived on Zenodo as Kawamura A (2024) Chd8_scRNA-seq: https://doi.org/10.5281/zenodo.19105439. The codes for the behavioral tests used in this study can be found on GitHub: https://github.com/daisukeino/BehavioralAnalysis, and archived on Zenodo as Ino D (2025) BehavioralAnalysis: https://doi.org/10.5281/zenodo.19105511.

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

## Acknowledgements

We thank M. Asamura, H. Kobayashi, M. Matsubara, and C. Tambo for overall technical assistance, D. Ino and other laboratory members for discussion, and K. Kaneda for cryopreserved sperm of *Vgat-ires-Cre* mice.

## Author contributions

K.N. and A.K. designed and performed most experiments, analyzed data, and prepared the manuscript. A.T. designed, performed, and analyzed the fiber photometry experiments and contributed to writing the photometry sections of the manuscript. T.I. contributed to the MEA recordings and to data interpretation and analysis. S.H., J.T. and T.D. contributed to the generation of *Chd8*[+/LSL] mice. K.H. and K. Kurokawa performed scRNA-seq analysis. K. Kato provided intellectual support. M.N. contributed to the design and supervision of the study and to writing the manuscript.

## Funding

K.N. was supported by a fellowship, Support for Pioneering Research Initiated by the Next Generation (SPRING) program, from Japan Science and Technology Agency (JPMJSP2135). A.K. was supported by a fellowship from the Japan Society for the Promotion of Science (JSPS, JP21J00911). This work was supported in part by KAKENHI grants from JSPS and the Ministry of Education, Culture, Sports, Science, and Technology of Japan to K.N. (JP25KJ0256, JP25K19075), A.K. (JP21K15726, JP21H05619, JP22H05493, JP23K14795, JP22H04925 (PAGS)), and to M.N. (JP21H02847, JP24H00627), by a PRIME grant from Japan Agency for Medical Research and Development (AMED) to M.N. (JP22gm6310008), by a Multidisciplinary Frontier Brain and Neuroscience Discoveries (Brain/MINDS 2.0) grant from AMED to M.N. (JP24wm0625316), by a grant from SECOM Science and Technology Foundation to M.N., and by a research grant from Astellas Foundation for Research on Metabolic Disorders to M.N.

## Competing interests

The authors declare no competing interests.
