## [Transparent Peer Review file · Nature Communications]

Defective ventral neurogenesis due to midfetal Chd8 mutation drives autistic-like behavior in mice

Corresponding Author: Professor Masaaki Nishiyama

Version 0:

Reviewer comments:

Reviewer #1

(Remarks to the Author)

The paper by Nitahara et al. identifies a critical period (midfoetal, E14.5) and a key cell population (ventral progenitor cells) that contribute to these behavioral deficits. Using lineage tracing, scRNA-seq, and spatial transcriptomics, they demonstrate that Chd8 haploinsufficiency alters the differentiation and gene expression patterns of inhibitory neurons and oligodendrocytes. Genetic rescue at the identified time point ameliorates behavioral and cellular phenotypes. Overall this is an excellent paper, methodologically sound with a tour-de-force of techniques. The multi-omics approach is commendable, and the rescue experiments strengthen the causal interpretation. The paper reports significant results (developmentally critical window), which should be of great interest to the broader N. Comms readership.

I have the following comments:

1. Behavioral analysis. 3-chamber test: did the experiment include a "counterbalance" empty wire cage in the non-social stimulus chamber? Direct social interaction: did the authors observe aggression in this test? While the authors examine social interaction and anxiety, they do not measure cognitive flexibility, motor stereotypies, or sensory processing, which are crucial ASD-related domains.
2. Are there any already published human scRNAseq data that could corroborate the findings in the Chd8 mouse model?
3. Excitatory neurons. Although the study focuses on inhibitory neurons and oligodendrocytes, it does not fully investigate how Chd8 loss affects excitatory neurons. Given that ASD involves excitatory/inhibitory imbalance, assessing synaptic activity (e.g., mEPSCs/mIPSCs) in cortical pyramidal neurons would have strengthened the mechanistic interpretation.
4. The study links Chd8 mutation to neurogenesis defects and behaviors reminiscent of ASD but lacks direct evidence of altered neuronal circuit function (apart from the NeuronChat analysis). Without electrophysiological recordings (e.g., patch-clamp, calcium imaging) or in vivo functional assays, the claim that GABAergic dysfunction underlies behavioral changes remains correlational rather than causative.

Reviewer #2

(Remarks to the Author)

The manuscript by Nitahara et al. shows that midfetal Chd8 mutation disrupts ventral neurogenesis, leading to autistic-like behaviors in mice, and identifies the critical period (between E14.5 and E17.5) for these phenotypes. The genetic rescue experiments are well-designed and provide valuable insights into the reversibility of behavioral abnormalities. However, some issues need to be addressed before the manuscript can be considered for publication.

Major comment,

#1. In the rescue experiments, shown in Figure 5 and Extended Figure 8, 9, 10, the authors claimed that the genetic rescue was successful based on the lack of a significant difference between the wild-type (Chd8^{+/+}) and the rescue group (Cre/Chd8^{+/-LSL}). The fact that the statistical test was not significant does not mean that there was no difference. Not being significant in the statistical test just means that we can't say that there is a difference. Direct statistical comparisons between the mutant group (Chd8^{+/-LSL}) and the rescue group (Cre/Chd8^{+/-LSL}) are missing, making it unclear whether the rescue truly reversed the mutant phenotype.

#2. Background strains and backcross counts are described for floxed mice, but not for LSL mice, Cre mice, etc. Please

provide these information.

#3. Multiple tests are performed on the behavioral analysis of mice. Are they done with naive mice each, or are multiple tests performed with the same mice?

If the same mice were used for different tests,

Minor comments,

#4. The manuscript refers to supp figs and tables as 1 to 3, while the extended figs are 1 to 10.

#5. Regarding code availability, the reviewer could not find any file on https://github.com/akwamura/Chd8_scRNA-seq. Codes for behavior analysis referred as "in-house software written in Python" should be also deposited.

Reviewer #3

(Remarks to the Author)

The manuscript titled "Defective ventral neurogenesis due to midfetal Chd8 mutation drives autistic-like behavior in mice" by Nitahara et al. elucidates the critical embryonic period responsible for behavioral abnormalities in adult Chd8 heterozygous mutated mice. Interestingly, the authors demonstrated that acceleration of differentiation of ventral progenitor cells causes autism-related behavior in adult mice. The experiments are well-designed, consisting of an impressive array of techniques to characterize the aberrant developmental program and the findings support a novel and critical pathogenesis due to Chd8 haploinsufficiency in neurodevelopment. The paper is generally well written and would be of interest to a broad scientific community. The authors are advised to address the following points to improve the manuscript.

Major concerns

1. (Fig. 3d-m, 2j) The data in Fig. 3d-m shows acceleration of cell-cycle exit in ventral progenitor cells in Chd8 heterozygous mice. Does the acceleration of cell cycle exit lead to the gene expression changes shown in Fig. 2j? More importantly, is it possible that the acceleration of cell cycle exit affects the cell proportions shown in Fig. 2h? These issues should be experimentally addressed or at least discussed.
2. (Fig. 4) Are the gene expression changes in Fig. 4 related to the gene expression changes in Fig. 2 or the acceleration of cell cycle exit shown in Fig. 3? Additionally, the results of interneuron expression changes should also be shown.
3. (Fig. 4m, n) Can the alterations in cell-to-cell communication in inhibitory neurons be explained by the results of the spatial transcriptome analysis? While RNA-seq might provide some insights, it alone would not be sufficient to fully explain the cellular phenotypes. These issues should be experimentally addressed or at least discussed.
4. (Fig. 5) For the development of therapeutic drugs, it would be better if treatment is possible even in adulthood. Does the expression of Chd8 in adulthood lead to the recovery of behavioral abnormalities? This issue should be experimentally addressed or at least discussed.

Minor concerns

1. (Fig. 1b) It seems to have fewer samples than the n=25 mentioned in the legend.
2. (Fig. 4) Which mice from Fig. 1a were used for this experiment, or are they global heterozygous mice?

Version 1:

Reviewer comments:

Reviewer #1

(Remarks to the Author)

The authors have done a substantial amount of new experiments and have addressed all the points raised in my review. The paper should be of great interest to the broader Nature Comms readership and the field of Neuroscience in particular.

Reviewer #2

(Remarks to the Author)

This manuscript has been well revised, and all technical issues now appear to have been resolved. The reviewer believes this manuscript is acceptable for publication.

Reviewer #3

(Remarks to the Author)

The authors have satisfactorily addressed all of the comments.

Response to Reviewer #1

The paper by Nitahara et al. identifies a critical period (midfoetal, E14.5) and a key cell population (ventral progenitor cells) that contribute to these behavioral deficits. Using lineage tracing, scRNA-seq, and spatial transcriptomics, they demonstrate that Chd8 haploinsufficiency alters the differentiation and gene expression patterns of inhibitory neurons and oligodendrocytes. Genetic rescue at the identified time point ameliorates behavioral and cellular phenotypes. Overall this is an excellent paper, methodologically sound with a tour-de-force of techniques. The multi-omics approach is commendable, and the rescue experiments strengthen the causal interpretation. The paper reports significant results (developmentally critical window), which should be of great interest to the broader N. Comms readership.

[Response] We thank the reviewer for the detailed and insightful comments on our manuscript, which we feel have helped us to greatly improve the paper, as well as for the statement that “The paper reports significant results (developmentally critical window), which should be of great interest to the broader N. Comms readership.”

1. Behavioral analysis. 3-chamber test: did the experiment include a "counterbalance" empty wire cage in the non-social stimulus chamber? Direct social interaction: did the authors observe aggression in this test? While the authors examine social interaction and anxiety, they do not measure cognitive flexibility, motor stereotypies, or sensory processing, which are crucial ASD-related domains.

[Response] We apologize for the incomplete description of the three-chamber tests. During the sociability phase (immediately after habituation), an identical wire cage was placed in the chamber opposite the cage containing the “stranger 1” mouse so as to provide a nonsocial stimulus. The right-left positions of the stranger-containing cage and the empty cage were systematically alternated between trials. We have now revised the Methods section of the manuscript to include these details (page 20, lines 655–657).

In response to the comment concerning aggression, we reexamined video recordings from the direct social-interaction test for evidence of aggression. We did not observe any aggressive behavior between the paired subject mice of either genotype

(*Nestin-Cre/Chd8^{+/+}* or *Nestin-Cre/Chd8^{+F}*), as has now been mentioned in the revised manuscript (page 4, lines 86–87).

As suggested by the reviewer, we have now also performed an attentional set-shifting task (ASST) test for cognitive flexibility. Following the established mouse ASST protocol, we quantified cognitive flexibility on the basis of the number of trials required to reach the criterion across intradimensional (IDS) and extradimensional (EDS) shifts. The *Chd8* mutant mice did not differ from the control mice with regard to IDS or EDS performance (new Supplementary Fig. 2a), indicative of no detectable impairment in cognitive flexibility. This result is consistent with our previous observation that ventral progenitor-specific mutation of *Chd8* was not associated with a cognitive deficit in the T-maze left-right discrimination test (Kawamura *et al.*, *Hum. Mol. Genet.* 2020).

In addition to the self-grooming test (Supplementary Fig. 1m), we quantified motor stereotypies by measuring perimeter circling, defined as continuous rotation along the outer wall, during an open-field session. The number of circlings divided by total distance traveled was similar for *Chd8* mutant and control mice (new Supplementary Fig. 2b). The results for perimeter circling and the grooming test indicate the absence of a genotype-dependent difference in motor stereotypies, consistent with our previous report (Katayama *et al.*, *Nature* 2016).

With regard to sensory processing, we evaluated somatosensory nociception with a 55°C hot-plate test. The time required for the first nocifensive behavior was again similar for *Chd8* mutant and control mice (new Supplementary Fig. 2c), consistent with our prior findings with the acoustic startle-response and hot-plate tests (Katayama *et al.*, *Nature* 2016; Kawamura *et al.*, *Hum. Mol. Genet.* 2020).

Nitahara *et al.* Supplementary Figure 2

2. Are there any already published human scRNAseq data that could corroborate the findings in the *Chd8* mouse model?

[Response] Yes. Several published human single-cell and single-nucleus RNA-seq studies support our findings with the *Chd8* mouse model. For postmortem human ASD cortex (frontal, parietal, and occipital regions), snRNA-seq analysis revealed cell type-specific and region-dependent alterations, with prominent changes observed in inhibitory neurons and oligodendrocyte precursor cells (Gandal *et al.*, *Nature* 2022).

In addition, scRNA-seq studies with human cerebral organoids found that CHD8 perturbation results in disrupted differentiation and consequent increased production of inhibitory neurons (Villa *et al.*, *Cell Reports* 2022; Paulsen *et al.*, *Nature* 2022). These organoid findings are consistent with our results for the *Chd8* mouse model and support a conserved role for CHD8 in the regulation of inhibitory neurogenesis across species.

We have now added these details to the Discussion of our revised manuscript (page 12, lines 369–375).

3. Excitatory neurons. Although the study focuses on inhibitory neurons and oligodendrocytes, it does not fully investigate how Chd8 loss affects excitatory neurons. Given that ASD involves excitatory/inhibitory imbalance, assessing synaptic activity (e.g., mEPSCs/mIPSCs) in cortical pyramidal neurons would have strengthened the mechanistic interpretation.

[Response] We agree on the importance of evaluating excitatory neurons. We previously performed whole-cell patch-clamp recordings from layer-2/3 pyramidal neurons in *Chd8* mutation mice and detected no significant EPSC/IPSC imbalance (Kawamura *et al.*, *Hum. Mol. Genet.* 2020). To complement these single-cell measurements, we recorded network-level activity with microelectrode arrays (MEAs) in primary cultures of E13.5 cortical neurons, which are predominantly excitatory (new Supplementary Fig. 10a–c). The mean firing rate was significantly reduced in *Chd8*^{+/-} (mutant) excitatory neurons, whereas spike amplitude and total axonal length were similar to those in control neurons (new Supplementary Fig. 10 d–h). These findings suggest that CHD8 haploinsufficiency results in a reduced excitability of excitatory neurons, consistent with previous evidence of molecular and functional abnormalities in excitatory neurons of *Chd8* mutant models (Ellingford *et al.*, *Mol. Psychiatry* 2021; Canales *et al.*, *bioRxiv* 2025). We have now described these MEA results in the revised manuscript (page 9, lines 282–290).

Nitahara *et al.* Supplementary Figure 10

4. The study links *Chd8* mutation to neurogenesis defects and behaviors reminiscent of ASD but lacks direct evidence of altered neuronal circuit function (apart from the NeuronChat analysis). Without electrophysiological recordings (e.g., patch-clamp, calcium imaging) or *in vivo* functional assays, the claim that GABAergic dysfunction underlies behavioral changes remains correlational rather than causative.

[Response] To address this point, we performed additional *in vivo* and *in vitro* functional analyses. To directly assess inhibitory circuit function *in vivo*, we combined fiber photometry with optogenetic manipulation in the medial prefrontal cortex (new Fig. 5). Using *Vgat-ires-Cre* mice, we selectively expressed the Ca²⁺ indicator

GCaMP6f and the red-shifted opsin ChrimsonR in inhibitory neurons and confirmed that optogenetic stimulation reliably evokes robust Ca^{2+} responses in VGAT-expressing interneurons (new Fig. 5a–e). We then examined the functional impact of inhibitory neurons on other neurons by expressing ChrimsonR in inhibitory neurons and GCaMP6f in surrounding neurons (new Fig. 5f–h). Optogenetic activation of inhibitory neurons induced a transient suppression on GCaMP6f-positive neuronal activity *in vivo*. Importantly, this light-evoked suppression was significantly attenuated in *Chd8* mutant (*Chd8*^{+/-}) mice compared with control (*Chd8*^{+/+}) mice (new Fig. 5i–k), providing direct evidence of impaired functional GABAergic connectivity *in vivo*.

To further characterize cell type-specific functional alterations, we performed MEA recordings. We confirmed that neurons in primary cultures of the E13.5 ganglionic eminence were predominantly inhibitory (new Supplementary Fig. 10b,c). The inhibitory neurons from *Chd8* mutant embryos had a significantly shorter axonal branch length compared with those from control mice, whereas firing frequency and spike amplitude were similar for both genotypes (new Supplementary Fig. 10i–m), suggesting that *Chd8* mutation diminishes the extent to which inhibitory neurons are able to influence their surrounding neural network.

Together, these new functional data demonstrate that CHD8 haploinsufficiency impairs inhibitory circuit connectivity *in vivo* and alters intrinsic functional properties of inhibitory neurons, thereby providing support for a mechanistic link to the observed behavioral phenotypes. We have now described the results of these *in vivo* and *in vitro* functional assays in the revised manuscript (page 8, lines 239–294).

Nitahara *et al.* Figure 5

Nitahara *et al.* Supplementary Figure 10

Response to Reviewer #2

The manuscript by Nitahara et al. shows that midfetal Chd8 mutation disrupts ventral neurogenesis, leading to autistic-like behaviors in mice, and identifies the critical period (between E14.5 and E17.5) for these phenotypes. The genetic rescue experiments are well-designed and provide valuable insights into the reversibility of behavioral abnormalities. However, some issues need to be addressed before the manuscript can be considered for publication.

[Response] We thank the reviewer for the careful review of our work and for the statement that “The genetic rescue experiments are well-designed and provide valuable insights into the reversibility of behavioral abnormalities.” We feel that the suggestions made have helped us to greatly improve our manuscript. Our specific responses to the points raised are as follows:

Major comment,

#1. In the rescue experiments, shown in Figure 5 and Extended Figure 8, 9, 10, the authors claimed that the genetic rescue was successful based on the lack of a significant difference between the wild-type ($Chd8^{+/+}$) and the rescue group ($Cre/Chd8^{+/LSL}$). The fact that the statistical test was not significant does not mean that there was no difference. Not being significant in the statistical test just means that we can't say that there is a difference. Direct statistical comparisons between the mutant group ($Chd8^{+/LSL}$) and the rescue group ($Cre/Chd8^{+/LSL}$) are missing, making it unclear whether the rescue truly reversed the mutant phenotype.

[Response] As suggested by the reviewer, we have now made statistical comparisons between the mutant ($Chd8^{+/LSL}$) and rescue ($Cre/Chd8^{+/LSL}$) groups (new Figure 6b–g, new Supplementary Figs. 12–15). Although some of the comparisons between these two groups were not statistically significant, partial reversal of autistic behavioral phenotypes was generally confirmed.

#2. Background strains and backcross counts are described for floxed mice, but not for LSL mice, Cre mice, etc. Please provide these information.

[Response] We apologize for the lack of description on this point. In our experiments, *Nestin-Cre*, *NestinCreER^{T2}*, *CAG-CreER*, *Olig1-Cre*, and *Rosa26-tdTomato* mice were backcrossed onto the C57BL/6J line for at least nine generations. For *Chd8* LSL mice, offspring were backcrossed onto the C57BL/6J background for at least five generations after CRISPR/Cas9-based genome editing in the same C57BL/6J strain. We have now included these details in the Methods section of the revised manuscript (page 14, lines 419–420 and 441–442).

*#3. Multiple tests are performed on the behavioral analysis of mice. Are they done with naive mice each, or are multiple tests performed with the same mice?
If the same mice were used for different tests.*

[Response] We again apologize for the insufficient description. The behavioral tests were conducted as a battery with the same cohort of mice in the following order: open-field test, light-dark transition test, elevated plus-maze test, sociability and social-novelty tests, social-interaction test, and self-grooming test. The hot-plate test and cognitive flexibility test were newly performed sequentially with another cohort of mice. Each test was separated by an interval of at least 24 h. We have now clarified this point in the revised manuscript (page 20, lines 626–631).

• *Minor comments,*

#4. The manuscript refers to supp figs and tables as 1 to 3, while the extended figs are 1 to 10.

[Response] The revised manuscript now contains 15 supplementary figures and 3 supplementary tables, without extended figures.

*#5. Regarding code availability, the reviewer could not find any file on https://github.com/akwamura/Chd8_scRNA-seq
Codes for behavior analysis referred as “in-house software written in Python” should be also deposited.*

[Response] We apologize for the data unavailability for the codes including in-house software written in Python. We have deposited code data for the scRNA-seq and spatial transcriptome analyses at https://github.com/akwamura/Chd8_scRNA-seq, as well as codes for behavioral tests at <https://github.com/daisukeino/BehavioralAnalysis>. This information is now included in the revised manuscript (page 25, lines 799–802).

Response to Reviewer #3

The manuscript titled "Defective ventral neurogenesis due to midfetal Chd8 mutation drives autistic-like behavior in mice" by Nitahara et al. elucidates the critical embryonic period responsible for behavioral abnormalities in adult Chd8 heterozygous mutated mice. Interestingly, the authors demonstrated that acceleration of differentiation of ventral progenitor cells causes autism-related behavior in adult mice. The experiments are well-designed, consisting of an impressive array of techniques to characterize the aberrant developmental program and the findings support a novel and critical pathogenesis due to Chd8 haploinsufficiency in neurodevelopment. The paper is generally well written and would be of interest to a broad scientific community. The authors are advised to address the following points to improve the manuscript.

[Response] We thank the reviewer for the thorough evaluation of our manuscript and for the statement that “The paper is generally well written and would be of interest to a broad scientific community.” We feel that the comments have helped us to greatly improve our manuscript. Our specific responses to the points raised are as follows:

- *Major concerns*

1. (Fig. 3d-m, 2j) *The data in Fig. 3d-m shows acceleration of cell-cycle exit in ventral progenitor cells in Chd8 heterozygous mice. Does the acceleration of cell cycle exit lead to the gene expression changes shown in Fig. 2j? More importantly, is it possible that the acceleration of cell cycle exit affects the cell proportions shown in Fig. 2h? These issues should be experimentally addressed or at least discussed.*

[Response] The accelerated cell-cycle exit and differentiation observed in the ganglionic eminence of the *Chd8* mutant at E16.5 (Fig. 3d–m) are consistent with the altered transcriptional programs revealed by the scRNA-seq analysis in Figure 2. Specifically, GSEA indicated an overall enrichment of differentiation signatures in the inhibitory neuron, OPC, and oligodendrocyte clusters (Fig. 2i), and Figure 2j shows upregulation of genes that promote lineage progression, namely *Sox4* and *Sox11* (neuronal differentiation) in the inhibitory neuron cluster as well as *Nfia* and *Olig1* (oligodendrocyte differentiation) in the OPC cluster. Together, these data suggest that the accelerated cell-cycle exit and differentiation in the fetal GE likely contribute to the changes in gene expression detected in the scRNA-seq analysis. We are aware that these

datasets represent cross-sectional snapshots at different stages (E16.5 immunostaining vs. P5 scRNA-seq) and therefore do not provide direct causality, which we now state as a study limitation in the revised manuscript (page 12, lines 376–381; page 13, 395–398).

With regard to the proportions of cell types (Fig. 2h), the fractions of OPCs and mature oligodendrocytes were significantly higher in the *Chd8* mutant telencephalon, consistent with the notion that the accelerated differentiation of ventral progenitors results in excessive production of cells biased toward the oligodendrocyte lineage. By contrast, the inhibitory neuron fraction was unchanged. This difference may reflect lineage-specific developmental properties, given that OPCs remain proliferative during early postnatal stages, whereas inhibitory neurons are postmitotic and subject to postnatal programmed cell death (Lim *et al.*, *Neuron* 2018), which can normalize the earlier increase in their production. We have now addressed these points in our revised manuscript (page 5, lines 142–145; page 12, lines 381–385).

Nitahara *et al.* Figure 2

2. (Fig. 4) Are the gene expression changes in Fig. 4 related to the gene expression changes in Fig. 2 or the acceleration of cell cycle exit shown in Fig. 3? Additionally, the results of interneuron expression changes should also be shown.

[Response] In Figure 4, we observed downregulation of myelin-related genes and GABA signaling-related genes in cortical oligodendrocytes and interneurons of the

mutant, respectively, indicative of functional impairment of these cell types at the adult stage. In contrast, the analyses in Figures 2 and 3 revealed accelerated differentiation of these cell lineages in the P5 mutant mouse brain. This apparent discrepancy is likely attributable to the developmental stages at which these observations were made. Several models show that precocious cell-cycle exit of progenitors results in transient developmental advancement followed by hypofunction at later stages (Magno *et al.*, *Cell Rep.* 2021; Kaushik *et al.*, *Cell Rep.* 2025). In fact, previous research on *Chd8* heterozygous mutant mice found enhanced oligodendrocyte function up to P11 compared with wild-type controls, but reduced functionality at the adult stage (Jin *et al.*, *Science* 2020), consistent with our current findings. We have now included this information in our revised manuscript (page 13, lines 391–395).

With regard to the request to show interneuron expression changes, we have now added volcano plots for differentially expressed genes in the interneuron cluster of the mutant cortex and striatum compared with the control (new Supplementary Fig. 9a,b), with full statistics being provided in Supplementary Table 3. The expression of *Slc32a1*, which encodes a vesicular inhibitory amino acid transporter, was reduced in the interneuron cluster of the mutant cortex, consistent with diminished function of cortical interneurons at the adult stage. We have now addressed these results in the revised manuscript (page 8, lines 218–220).

Nitahara *et al.* Supplementary Figure 9

3. (Fig. 4m, n) Can the alterations in cell-to-cell communication in inhibitory neurons be explained by the results of the spatial transcriptome analysis? While RNA-seq might provide some insights, it alone would not be sufficient to fully explain the cellular phenotypes. These issues should be experimentally addressed or at least discussed.

[Response] To address this concern, we performed additional experimental analyses to directly assess inhibitory synaptic connectivity at both functional and structural levels. First, to directly evaluate the functional difference in inhibitory connectivity induced by *Chd8* mutation *in vivo*, we performed fiber photometry combined with optogenetic manipulation in the medial prefrontal cortex (new Fig. 5f). Optogenetic activation of inhibitory neurons induced a transient suppression of excitatory neuronal activity, and this inhibitory effect was significantly attenuated in *Chd8* mutant mice compared with control mice (new Fig. 5i–k). These results provide direct *in vivo* evidence that inhibitory neuron–mediated circuit function is compromised by CHD8 haploinsufficiency.

In addition, to examine inhibitory synaptic alterations from a structural perspective, we performed immunofluorescence staining for VGAT in the cortex. We found that the number of VGAT-positive puncta per soma was significantly reduced in the *Chd8* mutant cortex (new Fig. 5l,m), indicative of impaired inhibitory synapse formation. This finding is consistent with the downregulation of genes related to GABAergic synapse formation revealed by the spatial transcriptome analysis (Fig. 4g, new Supplementary Fig. 9a). Together, these data indicate that impaired inhibitory synaptic connectivity underlies the altered cell-to-cell communication in the *Chd8* mutant identified by the spatial transcriptome analysis. We have now addressed this issue in the Results section of the revised manuscript (page 8, lines 239–300).

Nitahara *et al.* Figure 5

4. (Fig. 5) For the development of therapeutic drugs, it would be better if treatment is possible even in adulthood. Does the expression of *Chd8* in adulthood lead to the recovery of behavioral abnormalities? This issue should be experimentally addressed or at least discussed.

[Response] As suggested by the reviewer, we examined whether restoration of CHD8 expression in adulthood might ameliorate behavioral phenotypes. To this end, we injected *CAG-CreER/Chd8^{+/-LSL}* mice with tamoxifen (100 mg/kg, i.p., for three consecutive days) starting at P20 and subjected them to behavioral tests at 9 to 13 weeks of age. Compared with *Chd8^{+/+}* mice, the *CAG-CreER/Chd8^{+/-LSL}* mice showed

abnormalities in social contact in the social-interaction test as well as increased anxiety-like behavior in the light-dark transition test, elevated plus-maze test, and open-field test, similar to *Chd8*^{+/-LSL} mice (Fig. R1, for reviewers only). Combined with our results showing that genetic rescue of *Chd8* expression at P1–3 similarly failed to normalize autistic-like behaviors (Fig. 6b–d), these new findings suggest that the restoration of *Chd8* expression in adulthood is unlikely to reverse the behavioral abnormalities associated with CHD8 haploinsufficiency.

Nitahara et al. Figure R1 for reviewers only

Minor concerns

1. (Fig. 1b) It seems to have fewer samples than the $n=25$ mentioned in the legend.

[Response] We apologize for the insufficient description of the sample size. The behavioral test battery in our study—including the open-field, light-dark transition, elevated plus-maze, and social-interaction tests—was conducted with 25 mice per genotype for each tamoxifen treatment time in Figure 1. In the social-interaction test, however, each pair of subject mice was counted as a single sample (one pair = two unfamiliar mice; $n = 12$ pairs per genotype). The sample number for the social-interaction test was therefore smaller than that for the other behavioral tests. We have now clarified this information in the revised manuscript (page 33, lines 1053–1055).

2. (Fig. 4) Which mice from Fig. 1a were used for this experiment, or are they global heterozygous mice?

[Response] In Figure 4, we used the adult brain of global *Chd8* heterozygous mutant (*Chd8*^{+/-}) and *Chd8* wild-type (*Chd8*^{+/+}) mice as mutant and control, respectively. We have now clarified this point in the Results and Methods sections of the revised manuscript (page 7, lines 202–203; page 39, lines 1115–1116).